# Verifier-free Test-Time Sampling for Vision-Language-Action Models

**Suhyeok Jang**[1]  **Dongyoung Kim**[1,3]  **Changyeon Kim**[1]  **Youngsuk Kim**[2]  **Jinwoo Shin**[1,3]

[1]KAIST  [2]Seoul National University  [3]RLWRLD

## Abstract

Vision-Language-Action models (VLAs) have demonstrated remarkable performance in robot control. However, they remain fundamentally limited in tasks that require high precision due to their single-inference paradigm. While test-time scaling approaches using external verifiers have shown promise, they require additional training and fail to generalize to unseen conditions. We propose Masking Distribution Guided Selection (MG-Select), a novel test-time scaling framework for VLAs that leverages the model's internal properties without requiring additional training or external modules. Our approach utilizes KL divergence from a reference action token distribution as a confidence metric for selecting the optimal action from multiple candidates. We introduce a reference distribution generated by the same VLA but with randomly masked states and language conditions as inputs, providing action uncertainty while remaining aligned with the target task distribution. Additionally, we propose a joint training strategy that enables the model to learn both conditional and unconditional distributions by applying dropout to state and language conditions, thereby further improving the quality of the reference distribution. Our experiments demonstrate that MG-Select provides a reliable reference for action selection through task-relevant condition masking and consistently improves base models across diverse simulation and real-world benchmarks.

## 1 Introduction

Vision-Language-Action models (VLAs; Zitkovich et al. 2023; Kim et al. 2024; Black et al. 2025; Bjorck et al. 2025), trained on large-scale robotic datasets (O'Neill et al., 2024; Bu et al., 2025), have demonstrated remarkable performance in robot control. Among these, autoregressive VLAs represent one of the predominant VLAs (Driess et al., 2023; Kim et al., 2024; Pertsch et al., 2025), leveraging the same autoregressive objective used in training vision and foundation models without requiring architectural modifications, yet achieving comparable performance to more sophisticated architectures. Despite their success, VLAs remain fundamentally limited in tasks that demand high precision; even after extensive pre-training, they often fail on fine-grained manipulation tasks such as grasping or object placement (Nakamoto et al., 2024; Kwok et al., 2025; Gu et al., 2025; Yang et al., 2025). This precision gap is particularly problematic for real-world robotic applications where millimeter-level accuracy can determine task success or failure.

Previous work (Nakamoto et al., 2024; Kwok et al., 2025) shows that while VLAs can achieve high precision with adequate training, their greedy decoding (always choosing the highest-probability action) becomes a bottleneck. To address this limitation, inspired by the substantial gains observed in LLM reasoning with Test Time Scaling (TTS) (Wang et al., 2023; Wan et al., 2025; Kang et al., 2025), they use repeated sampling paired with an external verifier, *i.e.*, a value function trained on robotic data. However, these approaches have significant drawbacks: First, they require additional training to obtain verifiers with reinforcement learning objectives before inference, which adds substantial computational overhead and complexity to the deployment pipeline. Second, these external verifiers fail to generalize to unseen input conditions (Nakamoto et al., 2024), such as novel task prompts or objects, and their reward modeling is tailored to specific datasets, severely limiting their broader applicability (Kwok et al., 2025).

**Our approach.** To tackle this problem, our research goal is to develop a test-time scaling framework for VLAs that leverages the model's internal properties without requiring additional training or

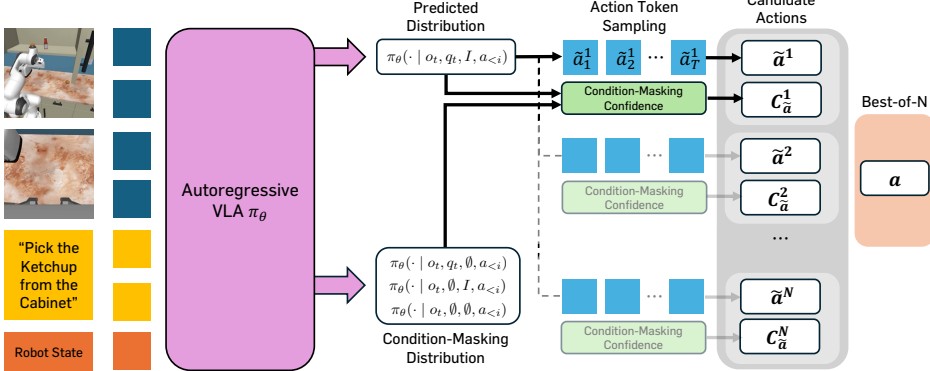

Figure 1: **Overview of MG-Select.** (1) Autoregressive VLA $\pi_\theta$ samples action tokens in parallel from the predicted distribution, while simultaneously computing token-wise KL divergence from the condition-masking distribution to the predicted distribution. (2) Best-of-N selection is then performed using an action confidence score $C_{\tilde{a}}$ obtained by aggregating these token-wise scores.

external modules. Inspired by verifier-free approaches for TTS (Zheng et al., 2024), we begin with the most straightforward approach: selecting the action with the highest likelihood from multiple sampled actions. We observe that this simple technique alone can improve VLA performance by producing more precise actions in some cases (see Table 5 (a)). However, this approach is not effective in general, as VLAs fine-tuned on target tasks for next action token prediction often memorize expert trajectories, causing the probability distribution over action tokens to become overly concentrated, which leads to multiple sampling converging to the same result.

These insights motivate us to propose Masking Distribution Guided Selection (MG-Select), a novel TTS framework that leverages the KL divergence from a reference action token distribution as a confidence metric for selecting the optimal action from multiple candidates. Inspired by recent advances in LLM literature that use self-certainty measures (Kang et al., 2025), we adapt this principle to the VLA setting. Specifically, we introduce a reference distribution generated by the same VLA but with randomly masked states and language conditions as inputs. This design ensures the reference distribution provides action uncertainty while remaining aligned with the target task distribution, yielding a more meaningful baseline for confidence measurement. By selecting actions with the highest KL divergence from this uncertainty-aware reference, MG-Select effectively identifies the most confident action sequences while avoiding the limitations of likelihood-based selection, achieving significant performance improvements in practice. Additionally, we propose a joint training strategy that enables the model to learn both conditional and unconditional distributions by applying dropout to state and language conditions, thereby further improving the quality of the reference distribution.

In our experiments, we have validated the effectiveness of our test-time scaling framework on both simulated (Nasiriany et al., 2024; Li et al., 2024; Liu et al., 2023) and real-world benchmarks (Khazatsky et al., 2024). Our results show that MG-Select consistently improves state-of-the-art VLAs (Pertsch et al., 2025) across diverse pick-and-place tasks and various environments. In particular, MG-Select achieves a **28%** improvement in real-world in-distribution tasks and **35%** in out-of-distribution tasks, along with a **168%** relative gain (5.3% → 14.2%) over vanilla greedy decoding on RoboCasa (Nasiriany et al., 2024) pick-and-place tasks trained with 30 demonstrations.

## 2 PRELIMINARIES

**Problem formulation.** We train the policy using the Imitation Learning (IL) framework. Specifically, IL formulates the robot control problem as a Markov Decision Process (MDP) (Sutton et al., 1998) without rewards, $\mathcal{M} = (\mathcal{S}, \mathcal{A}, P, \gamma, \rho_0)$, where $\mathcal{S}$ denotes the state space, $\mathcal{A}$ the action space, and $P(s' \mid s, a) \in [0, 1]$ is the transition probability from state $s \in \mathcal{S}$ to $s' \in \mathcal{S}$ given action $a \in \mathcal{A}$, $\gamma \in [0, 1)$ represents the discount factor, and $\rho_0$ denotes the initial state distribution. Given a policy $\pi_\theta$ and an expert demonstration dataset $\mathcal{D} = \{\mathcal{T}^i\}_{i=1}^{N_D}$, where each trajectory $\mathcal{T}^i = \{(s_t, a_t)\}_{t=1}^{|\mathcal{T}^i|}$ consists of state–action pairs of length $|\mathcal{T}^i|$, the policy is optimized such that $\pi_\theta(s_t)$ closely matches the expert action $a_t$ for each demonstration pair.

**Autoregressive VLA.** Given a state $s_t \in \mathcal{S}$ at timestep $t$ and an instruction $I \in \mathcal{L}$, we assume a language-conditioned VLA parameterized by $\theta$, $\pi_\theta : \mathcal{S} \times \mathcal{L} \to \Delta(\mathcal{A})$, where $\mathcal{L}$ denotes the space of possible language instructions and $\Delta(\mathcal{A})$ denotes the set of probability distributions over actions. The policy is trained on a demonstration dataset $\mathcal{D} = \{\mathcal{T}^i, I^i\}_{i=1}^{N_D}$ and outputs a distribution $\pi_\theta(a \mid s_t, I)$ over $a \in \mathcal{A}$. We further decompose the state into visual observation $o_t$ and proprioceptive state $q_t$, $s_t = (o_t, q_t)$ with $o_t \in \mathcal{O}$, $q_t \in \mathcal{Q}$, where $\mathcal{O}$ and $\mathcal{Q}$ denote the observation and proprioceptive state spaces, respectively. Therefore, the policy's action distribution can be expressed as $\pi_\theta(a \mid s_t, I) = \pi_\theta(a \mid o_t, q_t, I)$. In our test-time scaling framework, we utilize this distribution through repeated sampling to generate multiple candidate actions. In an autoregressive VLA, a continuous action chunk $a_{t:t+H}$ with time horizon $H$ is extracted from a trajectory $\mathcal{T}^i$ and tokenized into an action sequence $\mathbf{a} = (a_1, \ldots, a_T)$ of variable length $T$. The probability of an action sequence factorizes as

$$\pi_\theta(\mathbf{a} \mid o_t, q_t, I) = \prod_{k=1}^{T} \pi_\theta(a_k \mid o_t, q_t, I, a_{<k}),$$

where $a_{<k} = (a_1, \ldots, a_{k-1})$ is the prefix up to step $k-1$. Let $\mathcal{V}$ denote the vocabulary of discrete action tokens. At each step $k$, the model produces a logit vector $\ell_k \in \mathbb{R}^{|\mathcal{V}|}$ over $\mathcal{V}$. Applying the softmax function yields the next-token distribution $\pi_\theta(\cdot \mid o_t, q_t, I, a_{<k}) \in [0,1]^{|\mathcal{V}|}$, which is a categorical distribution over $|\mathcal{V}|$ possible tokens and sums to one.

## 3 METHOD

We present Masking Distribution Guided Selection (MG-Select), a novel test-time scaling framework that selects actions based on confidence scores from a reference action token distribution. In Section 3.1, we first introduce our overall test-time scaling framework. In Section 3.2, we introduce the confidence metric and its reference distribution used in our framework. In Section 3.3, we propose a joint training strategy for further improving the quality of the reference distribution in parallel with fine-tuning on the target dataset. We provide the overview of MG-Select in Figure 1. For additional details, please refer to Appendix A.

### 3.1 TEST-TIME SCALING FRAMEWORK

While VLAs demonstrate strong performance in robot control tasks, the single-inference paradigm becomes a bottleneck: the model always selects the most probable action from its predicted distribution (greedy decoding), even when this action may be suboptimal. This limitation is particularly problematic for tasks requiring high precision, such as fine-grained manipulation. To resolve this, we propose a test-time scaling framework that leverages only the model's internal signals, without relying on external verifiers. It consists of two stages: (1) parallel stochastic sampling to generate $N$ candidates, and (2) Best-of-N selection using a specific criterion $M$.

1. **Sampling $N$ candidate actions.** At timestep $t$, the autoregressive VLA $\pi_\theta$ samples actions $\mathbf{a} \in A$ from $\pi_\theta(\mathbf{a} \mid o_t, q_t, I)$. To obtain $N$ diverse candidates in parallel (batch-inference), we sample with temperature $\tau > 0$:

$$\tilde{a}_j^{(n)} \sim \pi_\theta(\cdot \mid o_t, q_t, I, \tilde{a}_{<j}^{(n)}; \tau), \quad n = 1, \ldots, N, \ j = 1, \ldots, T_n,$$

where $T_n$ denotes each action candidate's sequence length; $\pi_\theta(\cdot; \tau) = \mathrm{softmax}(\ell/\tau)$ controls distribution sharpness and sample diversity (close to greedy as $\tau \to 0$). This yields the candidate set $\tilde{A} = \{\tilde{\mathbf{a}}^{(n)}\}_{n=1}^{N}$ with $\tilde{\mathbf{a}}^{(n)} = (\tilde{a}_1^{(n)}, \ldots, \tilde{a}_{T_n}^{(n)})$.

2. **Best-of-N selection.** Among the $N$ candidate actions, we select the final action according to a pre-defined criterion $M$. This criterion is a metric for selecting the best candidate, and the selected action is given by:

$$\mathbf{a}^* = \underset{\tilde{\mathbf{a}}^{(n)} \in \tilde{A}}{\arg\max} \ M_{\tilde{\mathbf{a}}^{(n)}}.$$

### 3.2 CONDITION-MASKING DISTRIBUTIONAL CONFIDENCE FOR TEST-TIME SAMPLING

For test-time scaling, choosing a proper metric for selecting the best candidate is crucial for effectiveness. When multiple candidate actions are generated, we need a reliable way to identify the most

promising one. Intuitively, using the model's likelihood for action selection would be the simplest choice. However, this approach is not effective in general because VLAs fine-tuned on target tasks often produce overly concentrated probability distributions over action tokens, causing multiple sampling to converge to the same result. Instead, we propose a confidence metric based on the KL divergence between a predicted distribution and a reference distribution that represents uncertainty. This approach is motivated by the insight that actions that deviate most from an uncertainty-aware reference are likely to be the most confident and precise.

**Confidence over action token distributions.** We first define the action token distribution over the action vocabulary $\mathcal{V}$ as a probability distribution $P(a_i)$ where $a_i \in \mathcal{V}$ represents the $i$-th action token. While the VLA $\pi_\theta$ produces conditional distributions $\pi_\theta(\cdot \mid o_t, q_t, I, a_{<i})$ given observations, states, and instruction sequences, reference distributions can be constructed independently of such conditioning. These reference distributions can take various forms, such as uniform distributions over the action vocabulary, task-specific priors or other types of policy distributions. For computing the confidence over the action sequence, we first compute token-level distributional confidence at the $i$-th step token $a_i$ by measuring the distance between the predicted distribution $P_i = \pi_\theta(\cdot \mid o_t, q_t, I, a_{<i})$ and a reference distribution $Q_i$ as $C_i = \mathrm{KL}(Q_i \| P_i)$, where we use Kullback–Leibler (KL) divergence as our distributional confidence measure. We then aggregate these token-level confidences across the entire action sequence to obtain the final action-level confidence score for ranking candidate actions. Formally, for an action sequence $\mathbf{a} = (a_1, a_2, \ldots, a_T)$ of length $T$, we compute the action-level confidence as $C_{\mathbf{a}} = \sum_{i \in \mathcal{I}} C_i = \sum_{i \in \mathcal{I}} \mathrm{KL}(Q_i \| P_i)$, where $\mathcal{I} \subseteq \{1, 2, \ldots, T\}$ represents the set of token indices to be aggregated. The choice of $\mathcal{I}$ depends on the action tokenizing scheme: for full sequence aggregation, we use $\mathcal{I} = \{1, 2, \ldots, T\}$, while for partial aggregation, we select specific token ranges based on the tokenization structure.

**Condition-masking distribution.** To construct a reference distribution $Q$, our hypothesis is that a reference distribution that is uncertain yet not too distant from the target action token distribution will provide meaningful confidence signals. To this end, we mask specific information (Text, State, or both Text&State) from the input modalities given to the VLA $\pi_\theta$, creating condition-masking distributions that approximate failure modes where essential conditions for task solving are ignored. Formally, we compute the scoring metric as follows:

$$\textbf{(Text-masking)} \quad \mathrm{KL}_{\texttt{text}} = \mathrm{KL}\big(\pi_\theta(\cdot \mid o_t, q_t, \emptyset, a_{<i}) \,\|\, \pi_\theta(\cdot \mid o_t, q_t, I, a_{<i})\big), \tag{1}$$

$$\textbf{(State-masking)} \quad \mathrm{KL}_{\texttt{state}} = \mathrm{KL}\big(\pi_\theta(\cdot \mid o_t, \emptyset, I, a_{<i}) \,\|\, \pi_\theta(\cdot \mid o_t, q_t, I, a_{<i})\big), \tag{2}$$

$$\textbf{(Text\&State-masking)} \quad \mathrm{KL}_{\texttt{both}} = \mathrm{KL}\big(\pi_\theta(\cdot \mid o_t, \emptyset, \emptyset, a_{<i}) \,\|\, \pi_\theta(\cdot \mid o_t, q_t, I, a_{<i})\big), \tag{3}$$

For each task environment, the optimal confidence variant can vary. For example, in the SIMPLER-WidowX benchmark (Li et al., 2024), which consists solely of pick-and-place tasks, state-masking confidence works best because the model already memorizes how to pick and place objects without task instructions. In contrast, RoboCasa benchmark (Nasiriany et al., 2024), which has multiple task types, text-masking or text&state-masking are more effective, since the model cannot determine the correct action without instructions.

## 3.3 JOINT TRAINING STRATEGY

Although our method can be seamlessly integrated with any autoregressive VLA, existing VLAs are not trained under condition-masking settings, and directly masking inputs often leads to un-intended actions. To address this, we propose a new fine-tuning strategy that enables the model to generate condition-masking distributions while maintaining the performance gains from stan-dard fine-tuning on the target dataset. Specifically, we train the VLA with both all-condition and condition-masking data, randomly dropping certain conditions during fine-tuning to the target dataset, thereby increasing awareness of condition-masking distributions. Given the dataset $\mathcal{D}$, we augment it using four different masking variants applied to the proprioceptive state $q_t$ and the instruction $I$: $\mathcal{M} = \big\{(q_t, I), (q_t, \emptyset), (\emptyset, I), (\emptyset, \emptyset)\big\}$, corresponding to (i) all-condition, (ii) text-masking, (iii) state-masking, and (iv) both-masking cases. We then train the VLA with the augmented dataset $\mathcal{D}_{\text{augmented}}$ where $\mathcal{D}_{\text{augmented}} = \{(\mathcal{T}^i, I^i, q_t^{(m)}, I^{(m)}) \mid (q_t^{(m)}, I^{(m)}) \in \mathcal{M}\}$ as follows:

$$\mathcal{L}_{\text{Joint-IL}}(\theta; \mathcal{D}) = -\mathbb{E}_{((o_t, q_t), a_{t:t+H}, I) \sim \mathcal{D}} \left[ \mathbb{E}_{(q_t^{(m)}, I^{(m)}) \in \mathcal{M}} \left[ \log \pi_\theta(\mathbf{a}_t \mid o_t, q_t^{(m)}, I^{(m)}) \right] \right],$$

where $\mathbf{a}_t$ denotes the action sequence of length $T$ generated at timestep $t$. As a result, this fine-tuning strategy enables the VLA to maintain performance comparable to standard fine-tuning while gaining awareness of condition-masking distributions. When combined with our proposed confidence measure, this enhanced model (denoted as MG-Select*) demonstrates improved performance over the original Masking Distribution Guided Selection framework.

## 4 EXPERIMENTS

### 4.1 SIMULATED EXPERIMENTS

To validate the effectiveness of MG-Select, we conduct experiments across diverse robotic simulation environments including RoboCasa, SIMPLER-WidowX, and LIBERO. We fine-tune the pretrained $\pi_0$-FAST model for evaluation on all simulation environments, and additionally fine-tune OpenVLA for evaluation on LIBERO to demonstrate that our method improves performance regardless of the underlying model architecture.

### 4.1.1 SETUP

**RoboCasa (Nasiriany et al., 2024).** RoboCasa provides 24 atomic tasks set in household kitchen environments. We focus on 8 pick-and-place tasks, which are particularly challenging since they require high-precision actions (*i.e.*, grasping objects) and are well-suited for evaluating improvements in precision. Following Bjorck et al. (2025), we train the base model with 30, 100, and 300 demonstrations for each task. For comparison, we also report results for GR00T N1 (Bjorck et al., 2025), taken from the original paper.

**SIMPLER-WidowX (Li et al., 2024).** This benchmark evaluates whether our method improves precision in a real-to-sim setting. Because it does not provide simulated training data, we train the base model on BridgeData V2 (Walke et al., 2023) and evaluate it on 4 pick-and-place tasks. For comparison, we also report results for RT-1-X (O'Neill et al., 2024), Octo (Team et al., 2024), RoboVLM (Liu et al., 2025), and SpatialVLA (Qu et al., 2025), as reported in the SIMPLER paper (Li et al., 2024) and the respective original papers.

**LIBERO (Liu et al., 2023).** This benchmark evaluates multiple axes of generalization, including variations in layout, objects, and goals, as well as long-horizon tasks (LIBERO-Long) that require sustained high-precision actions.

### 4.1.2 EXPERIMENT RESULTS

**RoboCasa (Nasiriany et al., 2024).** Table 1 presents the performance of MG-Select with $\pi_0$-FAST (Pertsch et al., 2025) on RoboCasa. MG-Select consistently improves the base model across all tasks, including the 8 pick-and-place tasks, and under all demonstration scales. Notably, improvements appear even without joint training, showing that our test-time scaling alone can reliably select higher-precision actions. When combined with joint training, the gains are further amplified, since learning the condition-masking distribution during training provides a more reliable confidence signal for test-time scaling. We also observe particularly strong improvements in the low-data regime. For instance, with only 30 demonstrations, MG-Select with our joint training achieves a 168% relative improvement on pick-and-place tasks over the base model, highlighting that our method effectively compensates for limited performance under scarce data. We provide the detailed results in Appendix B.

**SIMPLER-WidowX (Li et al., 2024).** Table 2 shows the performance of MG-Select with $\pi_0$-FAST (Pertsch et al., 2025) on SIMPLER-WidowX. MG-Select clearly improves the base model across all tasks, demonstrating the robustness of our approach in enhancing action precision. We note that the base model performs relatively poorly on the "put eggplant in basket" task, since its background differs substantially from the other three tasks, making it sensitive to model-specific training configurations. For instance, SpatialVLA (Qu et al., 2025) achieves 100% success on the eggplant task but performs poorly on the remaining tasks. Despite this challenge, MG-Select still

Table 1: **Performance comparison on RoboCasa (Nasiriany et al., 2024).** We report the average success rate (%) over 50 trials on 24 tasks, including 8 pick-and-place tasks, trained with varying numbers of demonstrations. Results for our methods are averaged over 3 random seeds, while baseline results are taken as reported in the original paper. † indicates reproduced performance, and ∗ indicates results with additional joint training before applying our test-time scaling framework.

| Model | 30 Demos | | 100 Demos | | 300 Demos | |
|---|---|---|---|---|---|---|
| | Pick and Place | All | Pick and Place | All | Pick and Place | All |
| GR00T N1 | 0.4 | 17.4 | 2.2 | 32.1 | 22.6 | 49.6 |
| $\pi_0$-FAST† | 5.3 | 30.9 | 17.0 | 40.2 | 43.2 | 61.2 |
| + MG-Select (Ours) | 7.2 | 32.0 | 22.6 | 43.7 | 46.5 | 61.3 |
| **+ MG-Select* (Ours)** | **14.2** | **34.6** | **31.0** | **48.1** | **46.9** | **62.9** |

Table 2: **Performance comparison on SIMPLER-WidowX (Li et al., 2024).** We report the average success rate (%) over 24 trials on 4 pick-and-place tasks. Results for our methods are averaged over 3 random seeds, while baseline results are taken as reported in SIMPLER paper (Li et al., 2024) and the respective original papers. † indicates reproduced performance, and ∗ indicates results with additional joint training before applying our test-time scaling framework.

| Model | Spoon on Towel | Carrot on Plate | Stack Cubes | Eggplant in Basket | **Average** |
|---|---|---|---|---|---|
| RT-1-X | 0.0 | 4.2 | 0.0 | 0.0 | 1.1 |
| Octo | 12.5 | 8.3 | 0.0 | 43.1 | 16.0 |
| RoboVLM | 29.2 | 25.0 | 12.5 | 58.3 | 31.3 |
| SpatialVLA | 16.7 | 25.0 | 29.2 | **100.0** | 42.7 |
| $\pi_0$-FAST† | 66.7 | 70.8 | 41.7 | 8.3 | 46.9 |
| **+ MG-Select* (Ours)** | **69.4** | **75.0** | **43.1** | 13.9 | **50.3** |

provides consistent improvements on the eggplant task, indicating that our approach remains effective even when the base model struggles. For detailed results, please refer to Appendix B.

**LIBERO (Liu et al., 2023).** Table 6 presents the performance of MG-Select with $\pi_0$-FAST (Pertsch et al., 2025) on LIBERO. In this benchmark, we further extend our evaluation by applying MG-Select to OpenVLA (Kim et al., 2024), showing that our approach is compatible with different architectures. The results show that MG-Select achieves superior average performance over both base models, demonstrating its general effectiveness. Notably, LIBERO-Object and LIBERO-Long are the most challenging task suites (lowest base model performance), and the gains observed on these benchmarks highlight the effectiveness of our test-time scaling framework in improving precision. We provide further details about OpenVLA implementation in Appendix A.

## 4.2 REAL-WORLD EXPERIMENTS

To further validate our method's generalization beyond simulation environments, we conduct real-world robot experiments on a 7-DoF Franka Research 3 robot arm. We use the publicly released $\pi_0$-FAST-DROID as the base model for evaluation, which fine-tunes the pre-trained $\pi_0$-FAST on the DROID dataset (Khazatsky et al., 2024).

### 4.2.1 SETUP

**In-distribution tasks.** We design in-distribution (ID) tasks to evaluate the effectiveness of our method in enhancing base model performance under limited data, as task-specific real-world data is costly to collect. The ID tasks are pick-and-place tasks defined by a start and goal location, focusing on whether our method can generate high-precision actions for objects with different geometries, *e.g.*, a teddy bear, a cube, a rigid cup, and a sponge. For these tasks, we fine-tune the $\pi_0$-FAST-DROID model on 60 demonstrations per task, consisting of 15 demonstrations for each of the four objects.

**Out-of-distribution tasks.** We design out-of-distribution (OOD) tasks to evaluate whether our method improves the zero-shot generalization of the policy. We construct 2 OOD tasks involving unseen objects, *e.g.*, a lighter cup and a roll of tape. These OOD tasks are pick-and-place tasks similar to the ID tasks, but the policy must generalize to unseen real-world scenes and objects. The gains on these tasks reflect the effectiveness of our method in improving policy robustness.

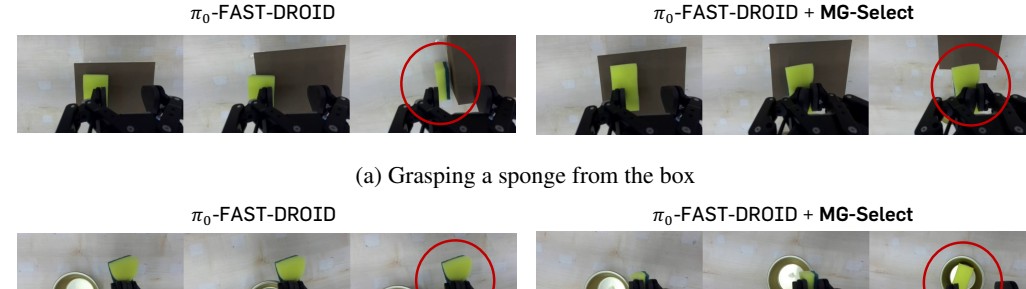

(a) Grasping a sponge from the box

(b) Releasing a sponge to the bowl

Figure 2: **Qualitative results of MG-Select in real-world pick-and-place tasks.** We visualize one of our real-world experiments in the "Box to Bowl" task: (a) grasping an object from the box and (b) releasing it into the bowl. The rollout shows that MG-Select can generate high-precision actions at critical moments for task success, whereas the base policy ($\pi_0$-FAST-DROID) often struggles at these steps.

Table 4: **Real-world performance on in-distribution tasks with Franka Research 3**. We evaluate our method on seen tasks after multi-task training with 60 demonstrations per task. Each task is defined by a start and goal location with 4 different target objects. We report the average success rate (%) over 24 trials (4 objects × 6 trials) for each task. ∗ indicates results with additional joint training before applying our test-time scaling framework.

| Model | Pick and Place | | | | Average |
|---|---|---|---|---|---|
| | Box to Bowl | Box to Plate | Basket to Bowl | Plate to Basket | |
| $\pi_0$-FAST-DROID | 41.7 | 37.5 | 45.8 | 25.0 | 37.5 |
| + MG-Select* (Ours) | **58.3** | **54.2** | **50.0** | **29.2** | **47.9** |

### 4.2.2 EXPERIMENTAL RESULTS

**In-distribution tasks.** Table 4 presents the performance of MG-Select on $\pi_0$-FAST-DROID (Pertsch et al., 2025) in in-distribution tasks. MG-Select outperforms the base model across all tasks, achieving a 28% relative gain. This demonstrates that our test-time scaling framework remains effective beyond simulation, enabling high-precision actions to complete pick-and-place tasks with diverse objects. We also provide qualitative results about real-world experiments in Figure 2, which show that MG-Select improves the precision of policy at critical moments of pick-and-place tasks, *i.e.*, grasping and releasing, where the base model often fails.

Table 3: **Real-world performance on out-of-distribution tasks with Franka Research 3**. We report the average success rate (%) over 16 trials for each task.

| Model | Pick up Tape | Take Cup out of Bowl | Average |
|---|---|---|---|
| $\pi_0$-FAST-DROID | 56.3 | 50.0 | 53.1 |
| + MG-Select (Ours) | **68.8** | **75.0** | **71.9** |

**Out-of-distribution tasks.** Table 3 presents the performance of MG-Select on $\pi_0$-FAST-DROID in out-of-distribution tasks. The results demonstrate that MG-Select can be directly applied to a generalizable policy, enhancing its robustness and precision on novel objects, with a 35% improvement. Notably, MG-Select shows clear gains on objects that are more difficult to grasp and lift than in-distribution ones, *e.g.*, a roll of tape.

### 4.3 ABLATION STUDIES AND ANALYSES

We investigate the effectiveness of the proposed components on RoboCasa and conduct the inference latency analysis on LIBERO-Object. For component-wise analysis, we use models trained on RoboCasa with 100 demonstrations, whereas for the latency analysis, we use models trained on LIBERO.

**Inference strategy.** Table 5 (a) shows that low-temperature sampling (*e.g.*, $\tau = 0.5$) already improves over greedy decoding on the jointly trained model. Even simple Best-of-N strategies, such as selecting

| $M$ | $N$ | PnP | All |
|---|---|---|---|
| Greedy | 1 | 28.5 | 42.7 |
| Sampling | 1 | 27.6 | 43.8 |
| Uniform KL | 4 | 30.0 | 46.5 |
| Likelihood | 4 | 30.5 | 46.8 |
| MG-Select | 4 | **31.0** | **48.1** |

(a) **Inference strategy**

| $N$ | PnP | All |
|---|---|---|
| 1 | 27.6 | 43.8 |
| 2 | 30.0 | 46.2 |
| 4 | 31.0 | 48.1 |
| 8 | 30.0 | 46.9 |
| 16 | 30.7 | 46.1 |
| 32 | 31.0 | 46.6 |
| 64 | **33.3** | **48.4** |

(b) **Number of candidates**

| Text | State | PnP | All |
|---|---|---|---|
| ✓ | ✗ | **31.0** | **48.1** |
| ✗ | ✓ | 30.1 | 46.7 |
| ✓ | ✓ | 29.7 | 46.3 |

(c) **Condition-masking variants**

| Joint-IL | MG-Select | PnP | All |
|---|---|---|---|
| ✗ | ✗ | 17.0 | 40.2 |
| ✗ | ✓ | 22.6 | 43.7 |
| ✓ | ✗ | 28.5 | 42.7 |
| ✓ | ✓ | **31.0** | **48.1** |

(d) **Effect of joint training**

| $\tau$ | PnP | All |
|---|---|---|
| 0.5 | 27.5 | 43.9 |
| 1.0 | 28.8 | 44.3 |
| 2.0 | 25.4 | 43.8 |
| 4.0 | **31.0** | **48.1** |
| 8.0 | 30.0 | 45.5 |

(e) **Regularization temperature**

| $\mathcal{I}$ | PnP | All |
|---|---|---|
| Sum | 26.1 | 44.5 |
| Avg | 24.7 | 44.7 |
| First 1 | 25.5 | 44.1 |
| First 3 | 27.1 | 45.5 |
| First 5 | **31.0** | **48.1** |
| First 7 | 29.2 | 46.3 |
| First 10 | 26.6 | 45.1 |

(f) **Aggregation strategy**

Table 5: **MG-Select ablation experiments.** We present a component-wise analysis of our proposed test-time scaling framework on RoboCasa (Nasiriany et al., 2024), trained with 100 demonstrations. We report the average success rate (%) over 50 trials and 3 random seeds. Temperature ($\tau$) for stochastic sampling is fixed to 0.5 across all experiments. PnP denotes the 8 pick-and-place tasks, and All denotes the full set of 24 tasks. Gray rows indicate the main results reported in Table 1.

actions by likelihood or KL divergence against a uniform reference distribution (Kang et al., 2025), yield further gains. Building on this, MG-Select achieves the strongest improvements, confirming that condition-masking distributional confidence provides a more effective uncertainty signal.

**Number of candidates.** Table 5 (b) shows that performance increases up to $N = 64$, but the improvement beyond $N = 4$ is marginal. Considering computational efficiency, we adopt $N = 4$ as the practical point that yields meaningful precision gains.

**Condition-masking variants.** Table 5 (c) presents the results of different masking variants after joint training. Text-masking achieves the best performance, while other variants remain competitive and outperform the uniform baseline (Kang et al., 2025).

**Effect of joint training.** Table 5 (d) shows the effect of combining our joint training strategy with MG-Select. Joint training alone already outperforms vanilla imitation learning, likely because condition-masking prevents the model from overfitting. Coupling it with MG-Select yields further gains over using MG-Select alone, confirming the effectiveness of the proposed strategy.

**Regularization temperature.** We empirically find that naively using the condition-masking distribution ($\tau = 1.0$) as a reference does not work well, as shown in Table 5 (e), compared to uniform-based KL divergence (Kang et al., 2025). It is possibly because condition-masking distribution may be "peaked" around certain action tokens, which undermines the purpose of distributional confidence by failing to consider the entire probability distribution. To address this issue, we apply an appropriate high temperature (*e.g.*, $\tau = 4.0$) to the condition-masking distribution, which regularizes its concentration and results in superior performance.

**Aggregation strategy.** Table 5 (f) shows that the aggregation strategy for action-level confidence is crucial for selecting high-precision actions. Intriguingly, the naive summation of token-level confidence performs the worst, while truncating to the first 5 tokens works best. We hypothesize that the results may be correlated with the nature of the FAST tokenizer (Pertsch et al., 2025), *i.e.*, each action sequence is composed of a variable number of action tokens, which are aligned from low- to high-frequency. We provide an additional analysis on the truncated FAST tokens and Robocasa performance in Appendix H.

Table 6: **Performance comparison on LIBERO (Liu et al., 2023)**. We report the average success rate (%) over 4 task suites, each consisting of 10 tasks with 50 trials per task. Results for our methods are averaged over 3 random seeds. † indicates reproduced performance, and ∗ indicates results with additional joint training before applying our test-time scaling framework.

| Model | LIBERO-Spatial | LIBERO-Object | LIBERO-Goal | LIBERO-Long | **Average** |
|---|---|---|---|---|---|
| OpenVLA† | **85.0** | 63.4 | **75.6** | 52.2 | 69.1 |
| **+ MG-Select* (Ours)** | 84.8 | **72.3** | 74.9 | **54.7** | **71.7** |
| $\pi_0$-FAST† | **97.4** | 95.4 | **95.6** | 79.6 | 92.0 |
| **+ MG-Select* (Ours)** | 97.2 | **98.0** | 94.5 | **82.7** | **93.1** |

**Effect of single prefill deployment.** Since MG-Select generates multiple candidate actions in parallel, it inevitably introduces additional latency, as the prefill step must be repeated $N$ times. This issue is particularly critical for VLAs, which require prefilling at every step when generating action sequences conditioned on the current observation. To address this, we design a *single-prefill* deployment strategy that shares one prefill across all $N$ candidates before decoding. This significantly reduces the computational overhead, as shown in Figure 3: with $N = 4$, our deployment achieves a 45% reduction in latency compared to vanilla MG-Select. As a result, the inference time of MG-Select remains comparable to that of single-action inference across different candidate sizes. We provide the detailed results in Appendix C.

## 5 RELATED WORK

**Vision-Language-Action models.** Developing generalist robot policies has long been a central objective in robotics. Recently, Vision-Language-Action models (VLAs) have emerged as a prominent approach, showing strong performance across diverse downstream tasks through large-scale pre-training on robotic datasets (Driess et al., 2023; Zitkovich et al., 2023; Black et al., 2025; Pertsch et al., 2025; Bjorck et al., 2025). Two common design paradigms have been explored: augmenting a vision-language model (VLM) with a diffusion-based action expert (Black et al., 2025; Bjorck et al., 2025), or converting the VLM into a VLA in an autoregressive manner (Kim et al., 2024; Pertsch et al., 2025). However, despite these advances, they fundamentally rely on a single-inference paradigm to generate actions, which increases the risk of errors in high-precision tasks.

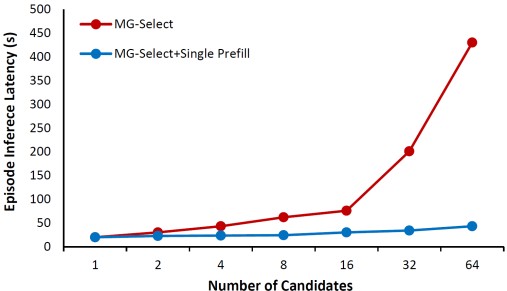

Figure 3: **Inference latency comparison on LIBERO-Object.** We compare vanilla MG-Select with its efficient deployment variant using single prefill, based on $\pi_0$-FAST (Pertsch et al., 2025).

**Test-time computing.** Applying additional computation at test time is widely recognized as an effective approach to generate more accurate outputs for challenging tasks across domains. In large language models (LLMs), numerous methods have demonstrated its effectiveness in improving reasoning capabilities, *e.g.*, mathematics, coding, and problem-solving (Chen et al., 2024; Brown et al., 2024; Ehrlich et al., 2025; Song et al., 2024). In robotics, test-time scaling has recently emerged as a promising paradigm, which involves repeated sampling combined with external value functions. For example, Nakamoto et al. (2024) ranks candidate actions using a value function trained via offline reinforcement learning on diverse robotic datasets, while Kwok et al. (2025) introduces VLM-based action verifiers obtained through reward modeling with synthetic preference datasets. Unlike these approaches, MG-Select requires no external modules. It performs Best-of-N sampling using only the model's intrinsic signals, *i.e.*, condition-masking distributional confidence. MG-Select consistently improves performance across diverse pick-and-place tasks. Moreover, it offers an efficient framework by eliminating the need for external model loading or interaction, and by introducing optimized parallel sampling that reduces inference latency.

## 6 CONCLUSION

In this work, we propose MG-Select, a novel test-time scaling framework for Vision-Language-Action models (VLAs). Our approach leverages condition-masking distributional confidence as

a self-generated signal for Best-of-N sampling, enabling precise action selection without external verifiers. This framework mitigates the precision issues inherent in single-inference paradigms and consistently improves policy performance across a wide range of simulation and real-world benchmarks. In addition, we introduce a joint training strategy and optimized implementation to further enhance both effectiveness and efficiency. We believe MG-Select opens up a verifier-free test-time scaling paradigm in VLAs, improving robustness and precision solely through the model itself.

## REPRODUCIBILITY STATEMENT

We provide implementation details about training and deployment in Appendix A.

## ACKNOWLEDGMENTS

This work was supported by Institute of Information & communications Technology Planning & Evaluation (IITP) grant funded by the Korea government (MSIT) (RS-2019-II190075, Artificial Intelligence Graduate School Program (KAIST); RS-2024-00509279, Global AI Frontier Lab; RS-2025-02653113, High-Performance Research AI Computing Infrastructure Support at the 2 PFLOPS Scale). We are grateful to RLWRLD Inc. for generously providing compute resources and support for the real-world experiments conducted in this work.

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

# A    IMPLEMENTATION DETAILS

## A.1    TRAINING ON SIMULATION DATA

**Imitation learning.** We use two representative autoregressive VLA policies as base models:

- $\pi_0$-FAST (Pertsch et al., 2025): It uses Paligemma-3B VLM (Beyer et al., 2024) as the backbone and is trained on 2 NVIDIA A100 GPUs with full fine-tuning from the pre-trained checkpoint. Common training configurations are fixed with the AdamW optimizer and a cosine decay schedule, with `warmup_steps` = 1,000, `peak_lr` = 2.5e-5, `decay_lr` = 2.5e-6, and `decay_steps` = 30,000. Training steps, global batch size, and action chunk horizon vary by dataset as shown in Table 7.

Table 7: Training setups of $\pi_0$-FAST for different simulation benchmarks.

| Configuration | RoboCasa | | | SIMPLER-WidowX | LIBERO |
|---|---|---|---|---|---|
| | 30 demos | 100 demos | 300 demos | | |
| Training steps | 3k | 5k | 20k | 10k | 10k |
| Global batch size | 64 | 64 | 64 | 64 | 32 |
| Action chunk horizon | 16 | 16 | 16 | 5 | 10 |

- OpenVLA (Kim et al., 2024): It uses Prismatic-7B VLM (Karamcheti et al., 2024) as the backbone and is trained on 2 NVIDIA A100 GPUs with LoRA fine-tuning ($r = 32$) from the pre-trained checkpoint. We use a global batch size of 32 for LIBERO (Liu et al., 2023), while other training configurations follow the official OpenVLA implementation. Note that, consistent with the OpenVLA configuration, we train the model separately on each LIBERO benchmark rather than performing multi-task fine-tuning.

**Joint imitation learning.** Joint imitation learning strictly follows the training configuration of the aforementioned imitation learning, differing only in the data configuration, as it incorporates condition-dropout data. We consider 3 variants of dropout data, (1) text-masking, (2) state-masking, and (3) both text&state-masking. For $\pi_0$-FAST, we randomly dropout 10%/10%/10% (text/state/both text&state) in RoboCasa and LIBERO, and only dropout 10% of state data in SIMPLER-WidowX. For OpenVLA, we apply a 10% dropout only to text condition since OpenVLA does not receive state input.

## A.2    TRAINING ON REAL WORLD DATA

**Imitation learning.** We use the official $\pi_0$-FAST-DROID (Pertsch et al., 2025) checkpoint as the base model for real-world experiments. We fine-tune it on our manually collected dataset using 2 NVIDIA A100 GPUs with full fine-tuning for 10k steps. Training is performed with a global batch size of 64 and an action chunk horizon of 10. Other configurations are fixed with the AdamW optimizer and a cosine decay schedule, with `warmup_steps` = 300, `peak_lr` = 1e-5, `decay_lr` = 1e-6, and `decay_steps` = 30,000.

**Joint imitation learning.** We follow the same training configuration as the above imitation learning setup, while additionally applying random dropout of 10%/10%/10% (text/state/both text&state).

## A.3    DEPLOYMENT

MG-Select's main hyperparameters are: (1) the sampling temperature $\tau$, (2) the number of candidate actions $N$, (3) the variant of condition-masking, and (4) the regularization temperature for the condition-masking distribution. We search for the optimal configuration on each dataset within the following ranges: $\tau \in \{0.1, 0.3, 0.5, 0.7, 1.0\}$, $N \in \{4, 8\}$, variants $\in \{\text{text}, \text{state}, \text{text}\&\text{state}\}$, and regularization temperature $\in \{4.0, 6.0, 8.0, 10.0, 12.0, 14.0, 16.0\}$, and report the best result for each policy.

For aggregating token-level confidence scores, we use the summation of the first 5 tokens by default in $\pi_0$-FAST. In contrast, for OpenVLA, we use the average score across the entire token sequence, since its output sequence length is fixed to the action dimension of the training data. Additionally, only the text-masking variant is applied in OpenVLA, as it does not take state input.

# B    DETAIL RESULTS ON SIMULATION EXPERIMENTS

Table 8: **Detailed performance comparison on RoboCasa (Nasiriany et al., 2024)**. We report the average success rate (%) over 50 trials, trained with varying numbers of demonstrations. For clarity, the 24 tasks are grouped into three categories: pick-and-place, open-and-close, and others. Results for our methods are averaged over 3 random seeds, while baseline results are taken from the original paper (Bjorck et al., 2025). † indicates reproduced performance, and ∗ indicates results with additional joint training before applying our test-time scaling framework.

| Model | 30 Demos | | | | 100 Demos | | | | 300 Demos | | | |
|---|---|---|---|---|---|---|---|---|---|---|---|---|
| | Pick and Place | Open and Close | Others | All | Pick and Place | Open and Close | Others | All | Pick and Place | Open and Close | Others | All |
| GR00T N1 | 0.4 | 26.0 | 26.0 | 17.4 | 2.2 | 52.8 | 43.5 | 32.1 | 22.6 | 68.3 | 60.0 | 49.6 |
| $\pi_0$-FAST† | 5.3 | 51.3 | 39.2 | 30.9 | 17.0 | 60.7 | 46.6 | 40.2 | 43.2 | 74.7 | **67.4** | 61.2 |
| + MG-Select (Ours) | 7.2 | **53.7** | 38.9 | 32.0 | 22.6 | 63.2 | 48.9 | 43.7 | 46.5 | 76.1 | 64.3 | 61.3 |
| + MG-Select* (Ours) | **14.2** | 53.2 | **39.7** | **34.6** | **31.0** | **67.3** | **50.1** | **48.1** | **46.9** | **81.0** | 64.9 | **62.9** |

Table 9: **Detailed performance comparison on SIMPLER-WidowX (Li et al., 2024)**. We report both the task success rate and the grasp success rate (%) over 24 trials on 4 pick-and-place tasks. Results for our methods are averaged over 3 random seeds, while baseline results are taken from the SIMPLER paper (Li et al., 2024) and the respective original papers (Liu et al., 2025; Qu et al., 2025). † indicates reproduced performance, and ∗ indicates results with additional joint training before applying our test-time scaling framework.

| Model | Spoon on Towel | | Carrot on Plate | | Stack Cubes | | Eggplant in Basket | | **Average** | |
|---|---|---|---|---|---|---|---|---|---|---|
| | Grasp | Success | Grasp | Success | Grasp | Success | Grasp | Success | Grasp | Success |
| RT-1-X | 16.7 | 0.0 | 20.8 | 4.2 | 8.3 | 0.0 | 0.0 | 0.0 | 11.5 | 1.1 |
| Octo | 34.7 | 12.5 | 52.8 | 8.3 | 31.9 | 0.0 | 66.7 | 43.1 | 46.5 | 16.0 |
| RoboVLM | 54.2 | 29.2 | 25.0 | 25.0 | 45.8 | 12.5 | 58.3 | 58.3 | 45.8 | 31.3 |
| SpatialVLA | 20.8 | 16.7 | 29.2 | 25.0 | 62.5 | 29.2 | **100.0** | **100.0** | 53.1 | 42.7 |
| $\pi_0$-FAST† | 83.3 | 66.7 | **83.3** | 70.8 | **91.7** | 41.7 | 8.3 | 8.3 | 66.7 | 46.9 |
| + MG-Select* (Ours) | **87.5** | **69.4** | **83.3** | **75.0** | 79.2 | **43.1** | 26.4 | 13.9 | **69.1** | **50.3** |

# C    DETAIL RESULTS ON EFFICIENT DEPLOYMENT STRATEGY

Table 10: **Detailed inference latency comparison on LIBERO-Object.** This table presents the detailed results corresponding to Figure 3, comparing vanilla MG-Select and MG-Select with the single prefill strategy. We report the average inference latency over 10 episodes for each of 5 random seeds, across different numbers of $N$ candidates. Latency is measured on an NVIDIA A100 GPU. ↓ indicates lower values are better.

| $N$ | Latency (s, ↓) | |
|---|---|---|
| | MG-Select | MG-Select + Single Prefill |
| 1 | 20.2 | 20.2 |
| 2 | 30.4 | **22.7** |
| 4 | 43.4 | **23.7** |
| 8 | 62.0 | **24.3** |
| 16 | 76.0 | **30.4** |
| 32 | 201.0 | **34.1** |
| 64 | 430.0 | **43.1** |

# D  ADDITIONAL EXPERIMENTS

## D.1  EFFECT OF DROPOUT RATIO

Table 11: **Ablation experiment on dropout ratios.** We present a dropout ratio analysis for joint imitation learning on RoboCasa (Nasiriany et al., 2024), trained with 100 demonstrations. We report the average success rate (%) over 50 trials and 3 random seeds. From left to right, the dropout ratio corresponds to text, state, and both text&state conditions. PnP denotes the 8 pick-and-place tasks, and All denotes the full set of 24 tasks. Blue rows indicate the main results reported in Table 1.

| Dropout Ratio (%) | PnP | All |
|---|---|---|
| 5/ 5/ 5 | 24.2 | 45.3 |
| 10/10/10 | **31.0** | **48.1** |
| 20/20/20 | 27.9 | 46.1 |

**Dropout ratios on RoboCasa.** We investigate the effect of the dropout ratio in joint imitation learning for MG-Select. Table 11 shows that a ratio of 10%/10%/10% achieves the best performance. We hypothesize that small ratios (*e.g.*, 5%) are insufficient for learning a meaningful masking distribution, while large ratios (*e.g.*, 20%) make the masking distribution too similar to the all-condition distribution, leading our confidence metric to select suboptimal action.

## D.2  EFFECT OF AGGREGATION STRATEGY

Table 12: **Ablation experiment on aggregation strategies**. We report the average success rate (%) over 24 trials on 4 pick-and-place tasks in SIMPLER-WidowX (Li et al., 2024). Results for our methods are averaged over 3 random seeds. Blue rows indicate the main results reported in Table 2. † indicates reproduced performance, and ∗ indicates results with additional joint training before applying our test-time scaling framework.

| Model | Spoon on Towel | Carrot on Plate | Stack Cubes | Eggplant in Basket | **Average** |
|---|---|---|---|---|---|
| $\pi_0$-FAST[†] | 66.7 | 70.8 | 41.7 | 8.3 | 46.9 |
| **+ MG-Select* (First 1)** | 68.1 | 70.8 | **48.6** | 20.8 | **52.1** |
| **+ MG-Select* (First 5)** | **69.4** | 75.0 | 43.1 | 13.9 | 50.3 |
| **+ MG-Select* (First 10)** | 66.7 | **76.4** | 38.9 | **22.2** | 51.0 |
| **+ MG-Select* (Sum)** | 62.5 | 73.6 | 43.1 | 16.7 | 49.0 |
| **+ MG-Select* (Avg)** | 66.7 | 68.1 | 40.3 | 16.7 | 47.9 |

**Aggregation strategies on SIMPLER-WidowX.** We investigate whether the aggregation strategy for action-level confidence remains important in a different domain. Table 12 shows that the truncation strategy consistently outperforms naive summation and averaging, and intriguingly, truncating first 1 token yields the best performance. This suggests that domain-specific tuning of the token-span size can further improve MG-Select. Nevertheless, we adopt the first 5 tokens truncation as default, as it shows robust and superior performance across action domains.

## D.3 EFFECT OF MODEL SCALE

Table 13: **Additional performance comparison on LIBERO (Liu et al., 2023)**. We report the average success rate (%) over 4 task suites, each consisting of 10 tasks with 50 trials per task. Results for our methods are averaged over 3 random seeds. † indicates reproduced performance, and ∗ indicates results with additional joint training before applying our test-time scaling framework.

| Model | LIBERO-Spatial | LIBERO-Object | LIBERO-Goal | LIBERO-Long | **Average** |
|---|---|---|---|---|---|
| MiniVLA† | **79.4** | 38.8 | 68.0 | 30.2 | 54.1 |
| **+ MG-Select* (Ours)** | 76.8 | **60.9** | **72.1** | **32.3** | **60.5** |

**Implementation Details.** MiniVLA (Belkhale & Sadigh, 2024) is a 7× smaller variant of OpenVLA, containing only 1B parameters. It uses a Qwen 2.5 0.5B backbone while retaining the same ViT used in OpenVLA. We fine-tune MiniVLA with vanilla imitation learning on the full set of LIBERO training data for 30k iterations with a global batch size of 128. Joint imitation learning includes 10% text-only dropout, as MiniVLA does not take state inputs. We adopt the average aggregation strategy, because the output token length is fixed to the action dimension.

**Experimental Results.** Table 13 presents the performance of MG-Select with MiniVLA (Belkhale & Sadigh, 2024) on LIBERO. The results show MG-Select significantly outperforms the base model on average, indicating that our test-time scaling framework generalizes across different VLAs and model scales.

## D.4 COMPARISON WITH ADDITIONAL BASELINES

Table 14: **Additional performance comparison on SIMPLER-WidowX (Li et al., 2024)**. We report the average success rate (%) over 24 trials on 4 pick-and-place tasks. Results for RoboMonkey (Kwok et al., 2025) and our methods are averaged over 3 random seeds. Blue rows indicate the main results reported in Table 2. † indicates reproduced performance, and ∗ indicates models trained with joint imitation learning. For a fair comparison, we match the number of candidates in RoboMonkey to that used by MG-Select. Latency is measured on an NVIDIA RTX A6000 GPU.

| Model | $N$ | External Verifier | Latency (ms, ↓) | Spoon on Towel | Carrot on Plate | Stack Cubes | Eggplant in Basket | **Average** |
|---|---|---|---|---|---|---|---|---|
| $\pi_0$-FAST† | 1 | - | 616.2 (1.00×) | 66.7 | 70.8 | 41.7 | 8.3 | 46.9 |
| + RoboMonkey* | 4 | ✓ | 1194.9 (1.94×) | 68.1 | 72.2 | **44.4** | **18.1** | **50.7** |
| **+ MG-Select* (Ours)** | 4 | ✗ | 880.3 (1.43×) | **69.4** | **75.0** | 43.1 | 13.9 | 50.3 |

**Implementation Details.** RoboMonkey (Kwok et al., 2025) is a recent test-time scaling method that uses a LLaVA-7B VLM-based verifier to select the optimal action. We aim to investigate whether our confidence metric can perform as well as an external verifier. To explicitly compare the two approaches, we fix $\tau = 0.1$ and the $N = 4$ for parallel stochastic sampling, and remove the Gaussian perturbation and majority voting process in RoboMonkey, using only the verifier to perform Best-of-$N$ selection from sampled action chunks.

**Experimental Results.** Table 14 shows that MG-Select achieves competitive performance compared to RoboMonkey without requiring an external verifier. MG-Select is substantially more efficient than RoboMonkey in terms of latency, demonstrating that our method is not only effective but also a practical test-time scaling approach.

# E EFFECT OF TEMPERATURE IN CONDITION-MASKING DISTRIBUTION

In our proposed method, we utilize a condition-masking distribution to serve as the reference distribution for the confidence metric. Here, we define the general condition-masking distribution as $\pi_{\text{masked}}$, which corresponds to the first argument in the KL divergence terms (see Eq. (1), (2), and (3)). In this section, we provide a theoretical derivation how temperature $\tau$ affects the entropy of the reference distribution $\pi_{\text{masked}}$.

## E.1 THEOREM

**Theorem.** Let $\pi_{\text{masked}}(a; \tau)$ be the temperature-scaled condition-masking reference distribution. Then its entropy $H(\pi_{\text{masked}})$ is monotonically increasing in temperature $\tau > 0$:

$$\frac{\partial H(\pi_{\text{masked}})}{\partial \tau} \geq 0. \tag{4}$$

Thus, using a higher temperature (e.g., $\tau = 4.0$) explicitly increases the entropy of the reference distribution, preventing our confidence metric from being biased by the over-confidence (low-entropy) of the masked distribution itself.

## E.2 PROOF

Let $z(a)$ denote the logit (unnormalized log-probability) of an action $a \in \mathcal{A}$ from $\pi_{\text{masked}}$. The probability of $a$ under temperature $\tau$ is defined as:

$$\pi_{\text{masked}}(a; \tau) = \frac{\exp(z(a)/\tau)}{Z(\tau)}, \quad \text{where} \quad Z(\tau) = \sum_{a'} \exp(z(a')/\tau). \tag{5}$$

The entropy of this distribution is defined as:

$$H(\pi_{\text{masked}}) = -\sum_{a} \pi_{\text{masked}}(a; \tau) \log \pi_{\text{masked}}(a; \tau). \tag{6}$$

First, we expand the entropy term using the definition of the softmax distribution:

$$
\begin{aligned}
H(\pi_{\text{masked}}) &= -\sum_{a} \pi_{\text{masked}}(a; \tau) \left( \frac{z(a)}{\tau} - \log Z(\tau) \right) \\
&= \log Z(\tau) \sum_{a} \pi_{\text{masked}}(a; \tau) - \frac{1}{\tau} \sum_{a} \pi_{\text{masked}}(a; \tau) z(a) \\
&= \log Z(\tau) - \frac{1}{\tau} \mathbb{E}_{\pi_{\text{masked}}}[z(A)].
\end{aligned}
\tag{7}
$$

Differentiating $H(\pi_{\text{masked}})$ with respect to $\tau$. We first note the derivative of the log-partition function $\log Z(\tau)$:

$$
\begin{aligned}
\frac{\partial \log Z(\tau)}{\partial \tau} &= \frac{1}{Z(\tau)} \sum_{a} \exp\left( \frac{z(a)}{\tau} \right) \cdot \left( -\frac{z(a)}{\tau^2} \right) \\
&= -\frac{1}{\tau^2} \sum_{a} \pi_{\text{masked}}(a; \tau) z(a) \\
&= -\frac{1}{\tau^2} \mathbb{E}_{\pi_{\text{masked}}}[z(A)].
\end{aligned}
\tag{8}
$$

Differentiating Eq. (7) with respect to $\tau$:

$$\frac{\partial H}{\partial \tau} = \frac{\partial \log Z(\tau)}{\partial \tau} - \left( -\frac{1}{\tau^2} \mathbb{E}_{\pi_{\text{masked}}}[z(A)] + \frac{1}{\tau} \frac{\partial \mathbb{E}_{\pi_{\text{masked}}}[z(A)]}{\partial \tau} \right). \tag{9}$$

Substituting Eq. (8) into the first term, the expected value terms cancel out:

$$\frac{\partial H}{\partial \tau} = -\frac{1}{\tau} \frac{\partial \mathbb{E}_{\pi_{\text{masked}}}[z(A)]}{\partial \tau}. \tag{10}$$

Next, to evaluate the derivative of the expectation $\mathbb{E}_{\pi_{\text{masked}}}[z(A)]$, we use the chain rule for $\frac{\partial \pi_{\text{masked}}(a;\tau)}{\partial \tau}$:

$$\begin{aligned}
\frac{\partial \pi_{\text{masked}}(a;\tau)}{\partial \tau} &= \pi_{\text{masked}}(a;\tau) \frac{\partial \log \pi_{\text{masked}}(a;\tau)}{\partial \tau} \\
&= \pi_{\text{masked}}(a;\tau) \left( -\frac{z(a)}{\tau^2} - \frac{\partial \log Z(\tau)}{\partial \tau} \right) \\
&= \pi_{\text{masked}}(a;\tau) \frac{1}{\tau^2} \left( \mathbb{E}_{\pi_{\text{masked}}}[z(A)] - z(a) \right).
\end{aligned} \quad (11)$$

Using this, the derivative of the expectation becomes:

$$\begin{aligned}
\frac{\partial \mathbb{E}_{\pi_{\text{masked}}}[z(A)]}{\partial \tau} &= \sum_a z(a) \frac{\partial \pi_{\text{masked}}(a;\tau)}{\partial \tau} \\
&= \frac{1}{\tau^2} \sum_a \pi_{\text{masked}}(a;\tau) z(a) \left( \mathbb{E}_{\pi_{\text{masked}}}[z(A)] - z(a) \right) \\
&= \frac{1}{\tau^2} \left( (\mathbb{E}_{\pi_{\text{masked}}}[z(A)])^2 - \mathbb{E}_{\pi_{\text{masked}}}[(z(A))^2] \right) \\
&= -\frac{1}{\tau^2} \text{Var}_{\pi_{\text{masked}}}[z(A)].
\end{aligned} \quad (12)$$

Finally, substituting this result back into Eq. (10):

$$\frac{\partial H(\pi_{\text{masked}})}{\partial \tau} = -\frac{1}{\tau} \left( -\frac{1}{\tau^2} \text{Var}_{\pi_{\text{masked}}}[z(A)] \right) = \frac{1}{\tau^3} \text{Var}_{\pi_{\text{masked}}}[z(A)]. \quad (13)$$

Since $\text{Var}_{\pi_{\text{masked}}}[z(A)] \geq 0$ and $\tau > 0$, we conclude:

$$\frac{\partial H(\pi_{\text{masked}})}{\partial \tau} \geq 0. \quad (14)$$

## F  RESULTS ON CALVIN BENCHMARK

Table 15: **Performance comparison on CALVIN (Mees et al., 2022)**. We report the success rate for each instruction chain and the average number of consecutive successes over 5 instruction chains. The model is trained on environments A, B and C and zero-shot evaluation is performed on novel environment D. Results for our methods are averaged over 3 random seeds. † indicates reproduced performance, and ∗ indicates results with additional joint training before applying our test-time scaling framework.

| Method | Task | Tasks Completed in a Row (%) | | | | | Avg. Len (↑) |
|---|---|---|---|---|---|---|---|
| | | 1 | 2 | 3 | 4 | 5 | |
| $\pi_0$-FAST† | ABC → D | 96.0 | 85.8 | 74.4 | 62.4 | 50.6 | 3.69 |
| **+ MG-Select\* (Ours)** | ABC → D | **96.9** | **88.0** | **77.8** | **67.6** | **55.8** | **3.86** |

To demonstrate that our method is also effective for long-horizon, multi-step planning tasks, we evaluate MG-Select on CALVIN benchmark in the zero-shot setting.

**Setup.** CALVIN (Mees et al., 2022) consists of 34 distinct tasks and uses a Franka Panda Arm for manipulation. We evaluate on the ABC → D setting, measuring whether the model can execute long-horizon language-conditioned tasks in a zero-shot manner. We fine-tune $\pi_0$-FAST with vanilla imitation learning on environments A,B,C for 5k iterations with a global batch size of 32. Then, we evaluate the model on environment D using 1000 instructions. Joint imitation learning includes 10%/10%/10% dropout, and MG-Select searches for optimal deployment hyperparameters within the ranges defined in Appendix A.

**Experimental Results.** Table 15 presents that MG-Select consistently outperforms the base model across all consecutive tasks, demonstrating our proposed method's generalizability to long-horizon and multi-step planning.

## G   VISUALIZATIONS OF MG-SELECT

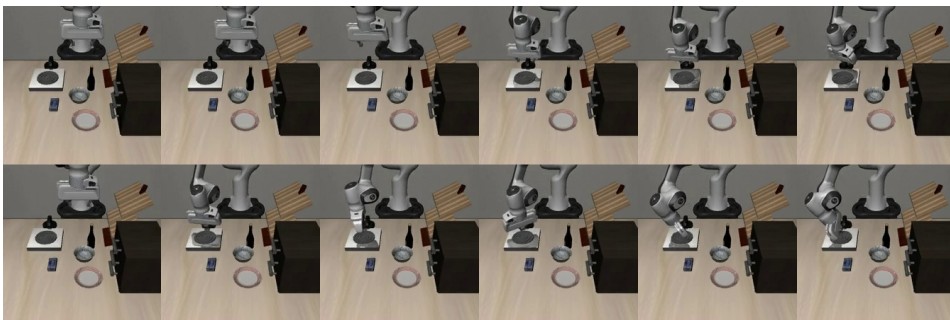

(a) Task : Turn on the stove. Top : OpenVLA, Bottom : OpenVLA + MG-Select (Ours).

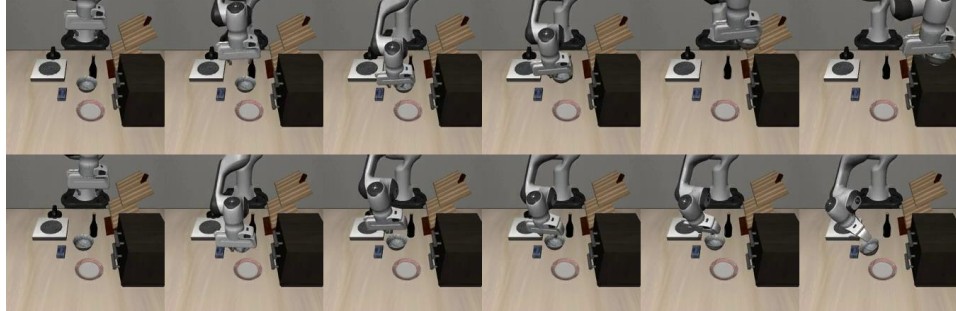

(b) Task : Put the bowl on top of the cabinet. Top : OpenVLA, Bottom : OpenVLA + MG-Select (Ours).

Figure 4: **Visualization of failure cases of MG-Select on LIBERO-Goal.** We show representative failures corresponding to minor performance drop in LIBERO-Goal. In simple and atomic tasks, MG-Select may introduce unnecessary stochasticity, leading to slight misalignment between the gripper and object, whereas the base model already produces near-optimal actions.

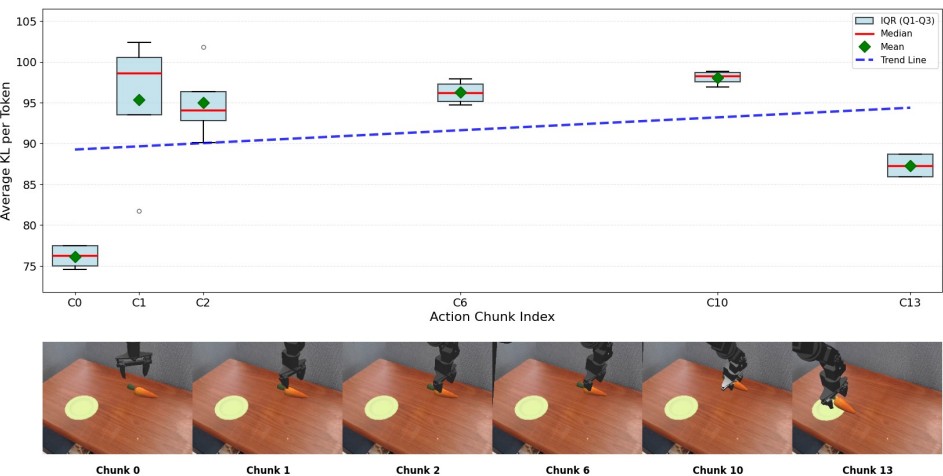

Figure 5: **Visualization of token-level KL divergence of MG-Select on SIMPLER-WidowX.** We visualize the token-level state-masking confidence (see Eq. (2)) for a successful pick-and-place episode in SIMPLER-WidowX. Average KL per token denotes the mean KL over first 5 action tokens, and the box-plots show KL statistics across action candidates ($N = 4$). Each frame corresponds to the state observed after executing the respective action chunk. We observe that KL rises sharply during the alignment phase (C1-C2), where state information is crucial for action prediction. At the same time, KL values among candidates also vary, and the highest-KL candidate produces the correct alignment. Similar patterns are obsersved in grasping (C10) and placement (C13).

## H    DETAIL ANALYSIS OF TRUNCATED FAST TOKENS

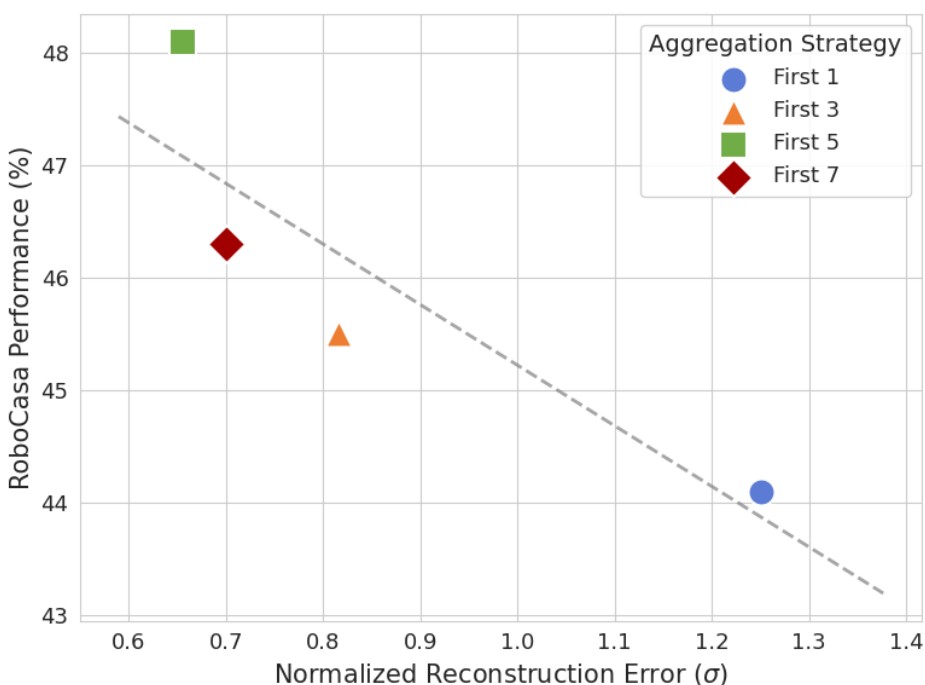

Figure 6: **Correlation between action reconstruction error and RoboCasa performance.** We compare the performance of truncated aggregation strategies (see Table 5 (f)) against the action reconstruction error using FAST (Pertsch et al., 2025) tokens on RoboCasa (Nasiriany et al., 2024) with 100 demonstrations. Reconstruction error is measured by tokenizing continuous actions and detokenizing them using only the first $K$ tokens. The error is reported as the Mean Absolute Error (MAE) normalized by the standard deviation of the original actions, averaged across all action dimensions.

**Action reconstruction from FAST tokens.** We analyze the behavior of the FAST (Pertsch et al., 2025) tokenizer, which produces variable-length token sequences ordered from low-frequency to high-frequency components. To understand how these tokens relate to action information, we conduct a simple action reconstruction experiment on the RoboCasa dataset using 100 demonstrations per task: (1) tokenize continuous actions into discrete FAST token sequences, (2) truncate the first $K$ tokens, and (3) detokenize the truncated sequence back into continuous actions. We evaluate $K \in \{1, 3, 5, 7\}$, corresponding to RoboCasa's 7-dimensional action space. As a reconstruction metric, we measure the mean absolute deviation between the original and detokenized actions with the first $K$ tokens, normalized by original action's standard deviation. Interestingly, we find that reconstruction error is negatively correlated with RoboCasa performance. In particular, using only the first 5 tokens provides a reasonable reconstruction, suggesting a more stable and length-invariant confidence measure than naively aggregating all FAST tokens, as shown in Table 5 (f).

# I  LLM USAGE DISCLOSURE

We acknowledge the use of large language models (LLMs) in preparing this manuscript. LLMs were employed solely to refine writing quality, including grammar correction, vocabulary suggestions, and typographical checks. All substantive ideas, analyses, and conclusions in this paper are entirely the work of the authors.

