# OpenReview forum: "Verifier-free Test-Time Sampling for Vision-Language-Action Models"
_ICLR.cc/2026/Conference — ICLR 2026 Poster_

### Official Review · Reviewer_CKBD · 2025-10-16

**Soundness:** 3
**Presentation:** 2
**Contribution:** 3
**Rating:** 4
**Confidence:** 3

**Summary:**

The paper proposes Masking Distribution Guided Selection (MG-Select), a verifier-free test-time scaling framework for Vision-Language-Action (VLA) models that enhances robotic action precision without additional training or external verifiers. MG-Select computes a confidence score based on the KL divergence between the model’s predicted action-token distribution and a reference distribution generated by masking input conditions (language or state), allowing the model to identify the most reliable action among multiple sampled candidates. A joint imitation learning strategy further improves this reference distribution by randomly dropping conditions during fine-tuning. Experiments on RoboCasa, SIMPLER-WidowX, LIBERO, and real-world Franka tasks show consistent gains—up to 168% relative improvement in low-data regimes—demonstrating that MG-Select effectively boosts precision and robustness while remaining computationally efficient and generalizable.

**Strengths:**

1. The paper proposes Masking Distribution Guided Selection (MG-Select), a verifier-free test-time scaling approach for Vision-Language-Action (VLA) models. The method leverages internal condition-masking distributions to construct a self-generated confidence metric, which is a timely and meaningful contribution to the emerging direction of verifier-free test-time optimization.

2. The methodology section (Section 3) is well structured and logically divided into sampling, confidence estimation, and joint training components.

3. The experiments are comprehensive, spanning multiple simulation benchmarks (RoboCasa, SIMPLER-WidowX, LIBERO) and real-world setups (Franka Research 3). The results demonstrate consistent improvements, indicating good generality.

4. The comparisons include strong baselines such as RT-1-X, Octo, RoboVLM, and SpatialVLA, which enhances credibility.

5. Table 5 provides a detailed ablation over candidate number, masking variant, and aggregation strategy.

6. The writing is clear, the structure follows ICLR standards, and Figure 1 effectively conveys the pipeline.

**Weaknesses:**

1. The boundary of innovation is somewhat vague. The proposed confidence estimation is closely related to recent “self-certainty” or “TrustScore” approaches in LLMs (e.g., Zheng et al., 2024). The paper does not clearly articulate how MG-Select differs conceptually or theoretically—whether it introduces a new mathematical formulation, or provides theoretical guarantees for its confidence metric.

2. Equation (2) for joint imitation learning is intuitively reasonable but lacks explanation regarding the fixed 10%/10%/10% masking ratio. No sensitivity analysis or theoretical rationale is provided for this choice. Similarly, the effect of the regularization temperature in the KL divergence is discussed empirically but not theoretically.

3. Despite extensive experiments, there is a lack of qualitative or failure-case analysis. The method is only evaluated on pick-and-place tasks, while scenarios requiring language reasoning or multi-step planning are absent.

4. Some baseline results are quoted from original papers, while MG-Select models are fine-tuned or re-trained under new settings, potentially introducing data or optimization biases.

5. The paper gives only empirical explanations for why aggregating the first five tokens yields the best performance, without analyzing how this relates to the FAST tokenizer structure or action semantics.

6. There are repeated phrases (e.g., “achieves significant performance improvements”), and the abstract’s statement of “168% relative gain” may be misleading without specifying the baseline reference.

**Questions:**

1. Include a theoretical justification of why the KL divergence from a masked reference distribution correlates with action optimality. Formal statements or lemmas demonstrating its reliability would strengthen the novelty claim.

2. Add more mathematical insight or justification in the appendix—for instance, derivations of how temperature regularization affects the entropy of the reference distribution, or an ablation on different masking ratios to validate robustness.

3. Include representative failure cases or visualizations to illustrate limitations (e.g., when MG-Select slightly underperforms in LIBERO-Goal). Consider adding evaluations on more complex, compositional manipulation tasks to assess generalization breadth.

4. Clearly specify which baselines are reproduced and which are cited from prior work. Provide full reproduction details (training steps, batch sizes, learning rates) in the appendix to ensure fair comparison.

5. Add a visualization of token-level KL divergence over time or provide statistics on token-wise confidence contributions to improve interpretability and transparency.

6. Tighten the abstract by focusing on conceptual insights rather than numeric claims, and clarify how relative improvements are computed. The conclusion could also better articulate MG-Select’s potential to shape future verifier-free test-time scaling paradigms in robotics.

---

> ### Author Response · Authors · 2025-11-22
> **Response to Reviewer CKBD (1/2)**
>
> Dear reviewer CKBD,
>
> We sincerely appreciate your efforts in reviewing our manuscript. We respond to each comment in the following content. In the revised manuscript, we have marked the revisions with “blue”.
>
> ---
> **[W1/Q1] How “MG-Select” differs conceptually or theoretically from the concept of “self-certainty” or “TrustScore” in LLMs?**
>
> MG-Select is specifically designed for VLAs, which operate on structured multi-modal inputs (e.g., observations, text and states), unlike LLMs. This multi-modal nature allows MG-Select to estimate uncertainty by masking non-visual conditions (e.g., text, state or both) while producing action distribution. Because the masking distribution preserves the visual context, it remains aligned with the target task distribution and enables the model to infer plausible task scenarios despite ambiguity about the exact action. We note that our confidence metric is closely connected to Mutual Information (MI) [1]. In particular, the conditional MI between the masked condition $M$ and the action $A$ given the unmasked condition $X$ is $I(M;A \mid X)= \mathbb{E}_{X,M}\left[{\mathrm{KL}}(p(A|X,M) \ || \ p(A|X))\right]$.  In our setting, the action confidence metric is computed at a specific conditioning pair $(x,m)$ as ${\mathrm{KL}}(p(a|x) \ || \ p(a|x,m))$, which serves as a point-wise estimator of the conditional MI, although in the reversed KL direction. Despite the reversed orientation, the metric similarly captures how strongly an action is influenced by the masked condition; selecting the action with the highest confidence naturally yields the most condition-aware action.
>
> [1] Cover et al., Elements of Information Theory, 2006
>
>
> ---
> **[W2/Q2] Lack explanation regarding the fixed masking ratio in joint imitation learning and the theoretical justification of regularization temperature in KL divergence**
>
> We first note that the fixed 10\%/10\%/10\% masking ratio in our joint training strategy is not arbitrary. It follows the prior work [1], which adopts a 10\% goal-condition dropout.  We additionally conduct an ablation on masking ratios, while keeping all other configurations as shown in the gray rows of Table 5. In the table below, the 10\%/10\%/10\% masking ratio achieves the best performance; we hypothesize that small ratios (5\%) are insufficient to learn the masking distribution, while large ratios (20\%) make the masking distribution too close to the full-condition distribution. Second, we clarify the role of the temperature $\tau$ in our masking distribution. Our confidence metric is defined as ${\mathrm{KL}}(p(A|X) \ || \ p(A|X,M))$, where $A$ denotes the action, $M$ the masked condition and $X$ the unmasked condition. This KL term is weighted by the condition-masking distribution $p(A|X)$. The temperature $\tau$ controls its sharpness: increasing $\tau$ smooths $p(A|X)$, preventing a few high-likelihood tokens from dominating the KL score. We have added the ablation results in Appendix D.1 and provided theoretical derivations on the effect of regularization temperature in Appendix E of our revised manuscript.
>
> \begin{array}{lcc} \hline \text{Masking Ratio (Text, State, Both)}  & \text{PnP} & \text{All} \newline \hline
> \phantom{0}5\\% / \phantom{0}5\\% / \phantom{0}5\\% & 24.2 & 45.3 \newline \hline
> 10\\% / 10\\% / 10\\% & \textbf{31.0} & \textbf{48.1} \newline \hline
> 20\\% / 20\\% / 20\\% & 27.9 & 46.1 \newline \hline \end{array}
>
>
>
> [1] Reuss et al., Goal-Conditioned Imitation Learning using Score-based Diffusion Policies, arXiv 2025

---

> ### Author Response · Authors · 2025-11-22
> **Response to Reviewer CKBD (2/2)**
>
> **[W3/Q3] Lack of qualitative or failure-case analysis and limited evaluation scope (only on pick-and-place tasks)**
>
> We clarify that MG-Select is a test-time scaling framework that is not limited to pick-and-place tasks. Nevertheless, to address your concern, we also demonstrate its effectiveness on CALVIN [1], a benchmark requiring multi-step planning with consecutive instruction chains evaluated in a zero-shot setting (see the table below). We have added  CALVIN results in Appendix F of our revised manuscript.
>
> In terms of failure cases, we believe that the minor performance drop in LIBERO-Goal is attributed to MG-Select introducing unnecessary stochasticity in simple, atomic tasks (e.g., “turn on the stove”), where the greedy action of the base model is already near-optimal. We have included failure-case visualizations for LIBERO-Goal in Appendix G of our revised manuscript. To mitigate the issue, we have also conducted additional hyperparameter tuning for MG-Select on OpenVLA for LIBERO-Spatial and LIBERO-Goal in Table 6 (with ranges specified in Appendix A.3), and updated the results in our revised manuscript; the overall trend remains consistent, while the performance drop of MG-Select is minimized.
>
> \begin{array}{lc|ccccc|c}
> \hline
> \phantom{Model} & \phantom{Task} & \rlap{~~~~~~\text{Tasks Completed in a Row}} & & & & & \phantom {Avg. Len (\uparrow)} \newline
> \hline
> \text{Model} & \text{Task} & \text{1} & \text{2}& \text{3} & \text{4} & \text{5} & \textbf{Avg. Len (\uparrow)} \newline
> \hline
> \text{$\pi_0$-FAST} & \text{ABC} \rightarrow \text{D} & \text{0.960} & \text{0.858} & \text{0.744} & \text{0.624} & \text{0.506} & \text{3.69} \newline
> \textbf{+ MG-Select (Ours)} & \text{ABC} \rightarrow \text{D} & \textbf{0.969} & \textbf{0.880} & \textbf{0.778} & \textbf{0.676} & \textbf{0.558} & \textbf{3.86} \newline
> \hline
> \end{array}
>
>
> [1] Mees et al., CALVIN: A Benchmark for Language-Conditioned Policy Learning for Long-Horizon Robot Manipulation Tasks, RAL 2022
>
> ---
> **[W4/Q4] Unclear method comparison; need clarification on which baselines are reproduced versus cited, and MG-Select models potentially introduce data or optimization biases**
>
> We clarify that all baselines in Tables 1 and 2 were included as references to enhance the reader’s understanding of performance comparisons and were directly cited from prior works. The effectiveness of MG-Select is consistently evaluated on top of the same base policy  $\pi_0$-FAST. We also clarify that MG-Select* models in Tables 1 and 2 do not introduce any data or optimization biases relative to $\pi_0$-FAST. This is because MG-Select* performs joint imitation training, while $\pi_0$-FAST uses vanilla imitation training under the exact same training configuration. Namely, MG-Select* differs from the baseline only in that it incorporates condition-dropout, as discussed in Appendix A of the original draft. Hence, we believe that our comparisons are fair.
>
> ---
> **[W5/Q5] Lack of theoretical or structural analysis on why aggregating the first five tokens performs best**
>
> Our first five tokens aggregation strategy is related to FAST [1] tokenizer’s ordering from low- to high frequency tokens, as discussed in Section 4.4 of the original draft. We observe that the first five tokens can reasonably reconstruct the underlying action; lower reconstruction error consistently corresponds to higher MG-Select performance. In contrast, aggregating KL scores over all tokens introduces a length bias, and simple averaging by token length does not resolve this issue (see “Sum” and “Avg” in Table 5f).  For your information, we have provided additional discussion on the relationship between action reconstruction and FAST token-span choices in Appendix H, along with a visualization of token-level KL divergence in Appendix G of our revised manuscript.
>
> [1] Pertsch et al., FAST: Efficient Action Tokenization for Vision-Language-Action Models, arXiv 2025
>
>
> ---
> **[W6/Q6] Editorial comments**
>
> We sincerely appreciate your constructive feedback. We have revised the draft to remove repeated phrases and to make the abstract more concept-focused. We clarified how relative improvements are computed in the introduction, and updated the conclusion to articulate MG-Select’s potential for verifier-free test-time scaling. For your information, we note that changed parts are colored in “blue” of our revised manuscript.

---

> > ### Comment · Reviewer_CKBD · 2025-11-23
> >
> > Thanks for your reply. Your messages address all of my concerns, and I will maintain my score.

---

> > > ### Author Response · Authors · 2025-11-23
> > > **Response to Reviewer CKBD**
> > >
> > > Dear reviewer CKBD,
> > >
> > > Thank you for your comments and for taking the time to review our manuscript. We are pleased to hear that our replies have resolved your concerns.
> > >
> > > If you have any further comments or remaining issues, please let us know. In particular, if there is a main concern that we should address for you to consider raising our score, we value your feedback and will provide a response to it.
> > >
> > > Thank you very much,\
> > > Authors

---

> > > > ### Comment · Reviewer_CKBD · 2025-11-25
> > > >
> > > > Thanks for your reply. Considering that your answer is very substantial, I have raised my score from 4 to 6.

---

### Official Review · Reviewer_6QZo · 2025-11-01

**Soundness:** 3
**Presentation:** 3
**Contribution:** 3
**Rating:** 6
**Confidence:** 3

**Summary:**

The paper proposes MG-Select, a self-verifying test-time selection mechanism for Vision-Language-Action (VLA) models. Instead of relying on an external verifier or value function, MG-Select computes KL divergence between masked and unmasked modality predictions as a measure of epistemic uncertainty. The approach enables verifier-free scaling at inference and is evaluated on multiple robot manipulation datasets (RoboCasa, LIBERO, DROID). The authors claim performance improvements on OOD (out-of-distribution) tasks while reducing verifier overhead.

**Strengths:**

1.	The paper identifies a key bottleneck in large multimodal agent systems: dependency on external verifiers during test-time scaling. To solve this, this paper attempts to replace it with a lightweight, self-contained mechanism. The idea of using internal divergence between masked/unmasked representations as a proxy for self-consistency is an intellectually interesting alternative to ensemble- or verifier-based uncertainty estimation. This motivation aligns well with emerging trends in verifier-free inference for embodied models like OpenVLA or RT-2.

2.	MG-Select’s implementation is refreshingly simple: sample multiple candidate actions, compute divergences under masked inputs, and select the candidate with the smallest divergence. The method avoids retraining or architectural modification, making it plug-and-play for deployed VLAs. This simplicity could enable fast adoption in practical robotic systems.

3.	The experiments span multiple datasets and include an ablation study on the number of samples and masking strategies. The consistency of improvement across different settings suggests the metric is robust to domain variation. Particularly, the evaluation on DROID with real-robot rollouts adds credibility to the claim of practical deployability.

**Weaknesses:**

1.	The KL divergence between masked and unmasked predictions is treated as an uncertainty signal, but the paper does not connect it to a formal epistemic uncertainty framework. For example, there’s no derivation relating it to Bayesian variance, entropy-based uncertainty, or self-consistency under dropout. Without such justification, it remains an empirical heuristic.

2.	While MG-Select eliminates external verifiers, it performs multiple forward passes (one per candidate), which could offset computational gains. The authors mention this but provide no latency or throughput benchmarks. In real-time control, inference latency often dominates, and without such metrics, claims of “efficiency” are weak.

3.	All evaluated tasks involve short-horizon manipulation with static visual backgrounds. The method’s efficacy under long-horizon planning, sequential reasoning, or language ambiguity (e.g., temporal grounding) remains unexplored.

**Questions:**

1.	Have you considered the effect of model scale—does MG-Select generalize across small and large VLAs?

2.	How sensitive is performance to the number of candidates (N)? Does uncertainty estimation saturate after a threshold?

3.	Could masked divergence be integrated into learning-time regularization to improve robustness, rather than only test-time selection?

---

> ### Author Response · Authors · 2025-11-22
> **Response to Reviewer 6QZo (1/2)**
>
> Dear reviewer 6QZo,
>
> We sincerely appreciate your efforts in reviewing our manuscript. We respond to each comment in the following content. In the revised manuscript, we have marked the revisions with “blue”.
>
> ---
> **[W1] KL divergence between masked and unmasked prediction appears as an empirical heuristic**
>
>
> We clarify that our confidence metric is not merely an empirical heuristic. Recent work [1] has justified using KL divergence from a uniform distribution to quantify model prediction confidence in LLMs. We adapt this principle to VLA by employing the condition-masking distribution as a more informative reference for uncertainty. We also note that our confidence metric is closely related to Mutual Information (MI) [2] in information theory. In particular, the conditional MI between the masked condition $M$ and the action $A$ given the unmasked condition $X$ is $I(M;A \mid X)= \mathbb{E}_{X,M}\left[{\mathrm{KL}}(p(A|X,M) \ || \ p(A|X))\right]$. In our setting, the action confidence metric is computed at a specific conditioning pair $(x,m)$ as ${\mathrm{KL}}(p(a|x) \ || \ p(a|x,m))$, which corresponds to a point-wise estimator of the conditional MI, although in the reversed KL direction. Despite the reversed orientation, our confidence metric still captures the reduction in action uncertainty due to the masked condition, providing a theoretical justification for our method.
>
> [1] Kang et al., Scalable best-of-n selection for large language models via self-certainty, arXiv 2025 \
> [2] Cover et al., Elements of Information Theory, 2006
>
> ---
> **[W2] Claims of “efficiency” are weak since no latency or throughput benchmarks are provided**
>
> We clarify that we already provide a latency benchmark for MG-Select in the original draft, where the inference latency across different numbers of candidates is reported in Appendix C, demonstrating the efficiency of our single-prefill strategy compared to the naive deployment strategy (see Section 4.4 of the original draft). MG-Select achieves significant improvements even with a small number of candidates, as shown in Table 5b, while latency increases by only 17\% at $N=4$ compared to single inference. For your information, we have expanded results on efficient deployment in Appendix C and revised the corresponding single-prefill section and Figure 3 of our revised manuscript.
>
> ---
> **[W3] Method’s efficacy in long-horizon planning, sequential reasoning remains unexplored**
>
> MG-Select is a test-time scaling framework that is not restricted to short-horizon manipulation tasks. Nevertheless, to address your concern, we also demonstrate its effectiveness on CALVIN [1], a benchmark requiring long-horizon planning with consecutive instruction chains evaluated in a zero-shot setting (see the table below), showing the generalizability of our proposed method. We have included the results in Appendix F of our revised manuscript.
>
> \begin{array}{lc|ccccc|c}
> \hline
> \phantom{Model} & \phantom{Task} & \rlap{~~~~~~\text{Tasks Completed in a Row}} & & & & & \phantom {Avg. Len (\uparrow)} \newline
> \hline
> \text{Model} & \text{Task} & \text{1} & \text{2}& \text{3} & \text{4} & \text{5} & \textbf{Avg. Len (\uparrow)} \newline
> \hline
> \text{$\pi_0$-FAST} & \text{ABC} \rightarrow \text{D} & \text{0.960} & \text{0.858} & \text{0.744} & \text{0.624} & \text{0.506} & \text{3.69} \newline
> \textbf{+ MG-Select (Ours)} & \text{ABC} \rightarrow \text{D} & \textbf{0.969} & \textbf{0.880} & \textbf{0.778} & \textbf{0.676} & \textbf{0.558} & \textbf{3.86} \newline
> \hline
> \end{array}
>
>
> [1] Mees et al., CALVIN: A Benchmark for Language-Conditioned Policy Learning for Long-Horizon Robot Manipulation Tasks, RAL 2022

---

> ### Author Response · Authors · 2025-11-22
> **Response to Reviewer 6QZo (2/2)**
>
> **[Q1] Does MG-Select generalize across small and large VLAs?**
>
> We kindly remark that we have demonstrated the effectiveness of MG-Select across different architectures and model sizes in Table 6 and Appendix A.1 of our original draft. In the table below, MG-Select also improves performance on an even smaller model, MiniVLA [1], which has approximately 1B parameters, indicating that MG-Select generalizes across VLAs of varying scales. We have added the results in Appendix D.3 of our revised manuscript.
>
> \begin{array}{lcccc} \hline \text{Model} & \text{LIBERO-Spatial} & \text{LIBERO-Object} & \text{LIBERO-Goal} & \text{LIBERO-Long} & \textbf{Average}  \newline \hline
> \text{MiniVLA [1]} & \textbf{79.4} & 38.8 & 68.0 & 30.2 & 54.1 &  \newline
> \textbf{+ MG-Select (Ours)} & 76.8 & \textbf{60.9} &  \textbf{72.1} & \textbf {32.3} & \textbf{60.5} \newline \hline \end{array}
>
>
> [1] Belkhale et al., MiniVLA: A Better VLA with a Smaller Footprint,    https://github.com/Stanford-ILIAD/openvla-mini
>
> ---
> **[Q2] How sensitive is performance to the number of candidates ($N$),  and does it saturate beyond a certain point?**
>
> We kindly remark that we have provided the ablation on the number of candidates in Table 5b of our original draft. The performance improves noticeably as $N$ increases from 1 to 4, showing that our uncertainty estimation is effective even with a small number of samples. We also observe that $N=4$ is a practical sweet spot; although performance increases slightly up to $N=64$, the improvement saturates. For your information, we have included extended ablation results in Table 5b of our revised manuscript.
>
> \begin{array}{ccc} \hline N  & \text{PnP} & \text{All} \newline \hline
> \phantom{0}1 & 27.6 & 43.8 \newline
> \phantom{0}2 & 30.0 & 46.2 \newline
> \phantom{0}4 & 31.0 & 48.1 \newline
> \phantom{0}8 & 30.0 & 46.9 \newline
> 16 & 30.7 & 46.1 \newline
> 32 & 31.0 & 46.6 \newline
> 64 & \textbf{33.3} & \textbf{48.4} \newline \hline
> \end{array}
>
> ---
> **[Q3] Can MG-Select be used in learning-time regularization?**
>
> We believe our confidence metric is applicable to learning-time regularization. As discussed in Section 3.3, the masking distribution is aligned with the target task distribution by joint training, and thus can serve as an informative reference for guiding condition-aware action. This KL term can be incorporated into the standard cross-entropy objective as an auxiliary loss by minimizing the negative KL divergence. We hypothesize that this encourages the model to better reflect the text or state condition, preventing the policy from being biased toward the observation.

---

> ### Author Response · Authors · 2025-11-26
> **Further Discussion Before the Deadline**
>
> Dear Reviewer 6QZo,
>
>
> Thank you once again for your time and thoughtful efforts in reviewing our paper.
>
>
> As the discussion period will conclude soon, we would like to gently remind you in case you have any remaining comments. We believe that we have sincerely and successfully addressed your concerns, supported by the corresponding additional experimental results.
>
>
> If you have any further concerns or questions, please feel free to let us know.
>
>
> Thank you very much,\
> Authors

---

> > ### Comment · Reviewer_6QZo · 2025-11-27
> >
> > I appreciate the authors' responses, and I maintain my score. Thank you and good luck.

---

### Official Review · Reviewer_wMxK · 2025-11-02

**Soundness:** 3
**Presentation:** 3
**Contribution:** 3
**Rating:** 6
**Confidence:** 4

**Summary:**

The paper proposes Masking Distribution Guided Selection (MG‑Select) to do test‑time scaling method for VLAs.

The idea is to generate N action candidates with stochastic decoding and then pick the one with the highest distributional confidence, defined as a KL divergence between the model’s conditional next‑action distribution and a reference distribution produced by the same model with certain inputs masked (text, state, or both). Experiments on RoboCasa, SIMPLER‑WidowX, LIBERO, and a Franka arm show consistent gains, especially in low‑data regimes.

**Strengths:**

Simple and practical: Uses only the base VLA; no external verifier or extra training objective at inference time. Easy to bolt on to existing autoregressive policies

Good performance gain especially in low-data and OOD settings.

**Weaknesses:**

This method adds some latency due to increased compute

**Questions:**

1. Why choose main baseline as pi-0-fast? Which is not the best performing model by itself. What if applied on other model such as OpenVLA-OFT
2. How does it compare to some test time training baseline such as simple entropy minimization methods?
3. Can you provide more details about the masking strategy and aggregation design.

---

> ### Author Response · Authors · 2025-11-22
> **Response to Reviewer wMxK**
>
> Dear reviewer wMxK,
>
> We sincerely appreciate your efforts in reviewing our manuscript. We respond to each comment in the following content. In the revised manuscript, we have marked the revisions with “blue”.
>
> ---
> **[W1] MG-Select adds some latency due to increased compute**
>
> We clarify that MG-Select introduces negligible latency when sampling multiple candidates. At $N=4$, MG-Select increases latency by only 17\% compared to single inference while providing substantial performance gains (see Appendix C and Table 5b). We note that our single-prefill strategy shares one prefill across all $N$ candidates, and therefore significantly reduces the computational overhead, as discussed in Section 4.4 of the original draft. For your information, we have expanded the results of efficient deployment in Appendix C and revised the corresponding section and Figure 3 of our revised manuscript.
>
>
> ---
> **[Q1] Why choose $\pi_0$-FAST as the main baseline? What if MG-Select is applied to other models, such as OpenVLA-OFT?**
>
> To the best of our knowledge, $\pi_0$-FAST is one of the strongest available autoregressive VLAs, and is the most commonly adopted baseline in recent VLA works [1, 2]. We note that OpenVLA-OFT integrates parallel decoding with action chunking and is no longer an autoregressive model; therefore, token-wise KL score cannot be computed directly. Instead, we validate the generality of MG-select by applying it to another VLA architecture, OpenVLA [3] (see Table 6), and observe consistent performance improvements.
>
> [1] Lee et al., MolmoAct: Action Reasoning Models that can Reason in Space, arXiv 2025 \
> [2] Huang et al., OTTER: A Vision-Language-Action Model with Text-Aware Visual Feature Extraction, ICML 2025 \
> [3] Kim et al., OpenVLA: An Open-Source Vision-Language-Action Model, CoRL 2024
>
> ---
> **[Q2] How does it compare to some test time training baseline, such as entropy minimization methods?**
>
>
> We believe that test-time training methods [1] are often less practical in VLA settings, because test-time optimization interferes with real-time execution in real-world environments, which is crucial for robotic manipulation. In contrast, MG-Select achieves strong performance without any parameter updates and introduces only minimal latency (see Figure 3). Nevertheless, our confidence metric could be incorporated into test-time training frameworks, for example, by maximizing the KL divergence from condition-masking distribution on test samples.
>
>
> [1] Lee et al., Tent: Fully Test-time Adaptation by Entropy Minimization, ICLR 2021
>
> ---
> **[Q3] Can you provide more details about the masking strategy and aggregation design?**
>
>
> First, we define our confidence metric via KL divergence from a reference action token distribution. We construct this reference distribution by randomly masking specific input conditions (e.g., text, state, or both), which removes task-relevant signals while preserving alignment with the target task distribution. Intuitively, an action that deviates more from this masked reference reflects the masked condition more strongly. Second, for the FAST [1] tokenization scheme, we design a truncation-based aggregation method for token-wise KL scores. Because FAST encodes continuous actions into variable-length token sequences, naively aggregating over all tokens is influenced by sequence length and leads to poor Best-of-N performance (see Table 5f). Since FAST encodes low-frequency components in the earliest tokens, and action can be reasonably reconstructed from these initial tokens, we compute confidence using only the first five token KL scores by default, which provides a stable, length-independent aggregation.
>
> [1] Pertsch et al., FAST: Efficient Action Tokenization for Vision-Language-Action Models, arXiv 2025

---

> ### Author Response · Authors · 2025-11-26
> **Further Discussion Before the Deadline**
>
> Dear Reviewer wMxK,
>
>
> Thank you once again for your time and thoughtful efforts in reviewing our paper.
>
>
> As the discussion period will conclude soon, we would like to gently remind you in case you have any remaining comments. We believe that we have sincerely and successfully addressed your concerns, supported by the corresponding additional experimental results.
>
>
> If you have any further concerns or questions, please feel free to let us know.
>
>
> Thank you very much,\
> Authors

---

> > ### Comment · Reviewer_wMxK · 2025-11-28
> >
> > Thank you for your response. I will maintain my scores.

---

### Official Review · Reviewer_jiih · 2025-11-03

**Soundness:** 2
**Presentation:** 3
**Contribution:** 3
**Rating:** 6
**Confidence:** 3

**Summary:**

This paper proposes MG-Select, a verifier-free test-time scaling framework for Vision-Language-Action (VLA) models in robotics. Instead of relying on external verifiers or further model training, MG-Select leverages a KL-based confidence metric between a conventional model distribution and a reference distribution generated by masking certain language or state inputs. The method incorporates a novel joint training strategy to improve representations for both conditional and unconditional inputs, and is validated through comprehensive experiments on simulation and real-world benchmarks, achieving substantial improvements in both in-distribution and out-of-distribution robotic tasks.

**Strengths:**

- Verifier-free paradigm: The work addresses a clear practical limitation of prior VLA test-time scaling methods by eschewing external verifiers and any additional model components, simplifying deployment.

- Elegant use of KL divergence: The use of KL divergence as a confidence measure between the “full information” and “condition-masked” distributions is both theoretically motivated and practically appealing, connecting well with recent LLM self-consistency work but tailored for the robotics domain.

**Weaknesses:**

1. While the masking variants (text, state, both) and truncation (first 5 tokens in Table 5f) offer empirical benefit, the rationale for these choices is left at the level of intuition/hypotheses. Given that “naive summation” (Table 5f) performs quite poorly, a more careful analysis—perhaps formalizing when/why these aggregation choices yield confident actions—should be pursued.

2.  The paper defines the action-level confidence as $C_{\mathbf{a}}=\sum_{i \in \mathcal{I}} \mathrm{KL}(P_i | Q_i)$ (Section 3.2), but is not explicit about (i) whether KL is computed at each timestep over the full action token vocabulary or just next-token distributions, (ii) how $\mathcal{I}$ is selected when token chunking is variable, and (iii) whether the reference $Q_i$ is static or recalculated for each sampled action sequence. The choices in Table 5f (Sum, Avg, First 5) are empirical, but the paper lacks a principled definition for $\mathcal{I}$ in general VLA architectures.

**Questions:**

1. Can the authors provide a direct, quantitative comparison or ablation with recent external-verifier methods (e.g., RoVer, RoboMonkey) on any of the main tasks or with open-sourced baselines, so as to substantiate claims of equal or better performance (and efficiency) of MG-Select?

2. Table 5 shows that “Text-masking” performs best for most tasks—can the authors formalize or empirically analyze why this is the case, and whether this result holds if tokenization schemes or action domains are changed? How robust is this preference for “text masking” versus other possible masking strategies?

3. The confidence metric $C_{\mathbf{a}}$ is calculated over the first 5 tokens (Table 5f). Is this choice robust? Does aggregation over different token span sizes appreciably affect the selection quality, or is this a dataset-specific artifact?

---

> ### Author Response · Authors · 2025-11-22
> **Response to Reviewer jiih (1/2)**
>
> Dear reviewer jiih,
>
> We sincerely appreciate your efforts in reviewing our manuscript. We respond to each comment in the following content. In the revised manuscript, we have marked the revisions with “blue”.
>
> ---
> **[W1] Why certain masking variants (text, state, both) and first 5 tokens truncation work better is not formally analyzed. When/Why does the proposed aggregation method yield confident actions?**
>
> We clarify that there is no universally optimal masking variant; the preferred choice depends on the action domain and task structure. For example, “text masking” is suitable for multi-task settings where different tasks can occur within identical scenes (e.g., RoboCasa). In contrast, “state masking” is preferable for similar pick-and-place tasks where proprioceptive information matters most (e.g., SIMPLER-WidowX). We note that our joint training framework flexibly supports all masking variants, and each yields consistent improvements over greedy decoding (see Table 5c). Second, we hypothesize that “naive summation” performs poorly because FAST [1] produces variable length sequences, causing action confidence to increase with sequence length rather than reflect true token-wise KL score. We observe that actions can be reasonably reconstructed from the initial tokens; aggregating only the early tokens (e.g., first 5 tokens) provides a more stable and length-invariant confidence estimate, as shown in Table 5f. For your information, we have included details on the relationship between action reconstruction and FAST token-span choices in Appendix H of our revised manuscript.
>
> [1] Pertsch et al., FAST: Efficient Action Tokenization for Vision-Language-Action Models, arXiv 2025
>
> ---
> **[W2-1] Whether KL is computed at each timestep over the full action token vocabulary or just next-token distributions?**
>
> We would like to clarify that our action-level confidence is computed on next-token distributions defined over the action token vocabulary, not over the full language model vocabulary. In practice, we precompute action token vocabulary from the training dataset of each benchmark.
>
>
> ---
> **[W2-2] How token indices are selected when token chunking is variable?**
>
> We clarify that the selection of token indices $\mathcal{I}$ is not constrained by token chunking variability. A naive summation aggregates token-wise KL scores over the entire sequence,  while averaging simply normalizes by the sequence length (denoted “Sum” and “Avg” in Table 5f), therefore both methods are affected by sequence length variability. A truncation strategy, such as the first 5 tokens selection, avoids this issue by consistently applying it regardless of the underlying token sequence length, using only the KL values of the first five tokens.
>
>
>
> ---
> **[W2-3] Whether the reference $Q_i$ is static or recalculated for each sampled action sequence?**
>
> We define the reference distribution $Q_i$, i.e., the condition-masking distribution, at every decoding step for each sampled action sequence. In particular, it is recalculated at each step using the previously sampled tokens from the predicted distribution $P_i$.

---

> ### Author Response · Authors · 2025-11-22
> **Response to Reviewer jiih (2/2)**
>
> **[Q1] Comparison with recent external-verifier methods (e.g., RoVer, RoboMonkey) ?**
>
> We compare MG-Select with a recent external-verifier method, RoboMonkey [1], on SIMPLER-WidowX benchmark in the table below. Our proposed method achieves competitive performance without an external verifier, whereas [1] requires a LLava-7B VLM verifier as a reward model for action selection. For a fair comparison, we match the number of action candidates ($N$) in [1] to that used by MG-Select. We have included these results in Appendix  D.4 of our revised manuscript.
>
>
> \begin{array}{lccc|cccc} \hline \text{Model} & N & \text{External Verifier} & \text{Latency (ms, \downarrow)} & \text{Spoon on Towel} & \text{Carrot on Plate} & \text{Stack Cubes} & \text{Eggplant in Basket} & \textbf{Average}  \newline \hline \text{$\pi_0$-FAST} & 1 & - & 616.2 \ (1.00 \times) & \text{66.7} & \text{70.8} & \text{41.7} & \text{8.3} & \text{46.9}  \newline \text{+ RoboMonkey [1]} & 4 & \checkmark & 1194.9 \ (1.94 \times) &\text{68.1} & \text{72.2} & \textbf{44.4} & \textbf{18.1} & \textbf{50.7} \newline \textbf{+ MG-Select (Ours)} & 4 & ✗ &  880.3 \ (1.43 \times) &  \textbf{69.4} & \textbf{75.0} & \text{43.1} & \text{13.9} & \text{50.3} \newline \hline \end{array}
>
>
>
> [1] Kwok et al., RoboMonkey: Scaling Test-Time Sampling and Verification for Vision-Language-Action Models, CoRL 2025
>
>
>
> ---
> **[Q2] Formalize or empirically analyze why “text-masking” performs best for most tasks, and whether this result holds if tokenization schemes or action domains are changed? How robust is the preference compared to other masking strategies?**
>
> First, we clarify that we do not claim the “text-masking” variant to be universally optimal across different action domains (see Section 3.2); the preferred masking strategy is task-dependent. We believe that text masking is particularly suitable for multi-task settings where faithfully following instructions is essential, especially when different tasks can occur within identical objects and scenes (e.g., RoboCasa). Second, we note that MG-Select is tokenizer-invariant. It performs well under a different tokenization scheme (see Table 6), i.e., discretizing each action dimension into one of 256 bins [1], which is used in OpenVLA.
>
> [1] Kim et al., OpenVLA: An Open-Source Vision-Language-Action Model, CoRL 2024
>
>
>
>
>
>
>
>
>
> ---
> **[Q3] Robustness of first 5 token aggregation is unclear; how does aggregation over different token-span sizes affect selection quality, and is the choice data-specific?**
>
> We clarify that MG-Select consistently improves performance across diverse simulation and real-world benchmarks on $\pi_0$-FAST, using the first 5 tokens aggregation strategy as the default (see Appendix C.3). We note that aggregation over different token span sizes does affect the selection quality, as shown in Table 5f. We also conduct an ablation on SIMPLER-WidowX, aggregating over the first single token yields the highest performance, indicating that the first 5 tokens aggregation is robust while additional data-specific tuning can further improve results. For your information, we have included extended ablation results in Table 5f and ablation results of SIMPLER-WidowX in Appendix D.2 in our revised manuscript.

---

> ### Author Response · Authors · 2025-11-26
> **Further Discussion Before the Deadline**
>
> Dear Reviewer jiih,
>
>
> Thank you once again for your time and thoughtful efforts in reviewing our paper.
>
>
> As the discussion period will conclude soon, we would like to gently remind you in case you have any remaining comments. We believe that we have sincerely and successfully addressed your concerns, supported by the corresponding additional experimental results.
>
>
> If you have any further concerns or questions, please feel free to let us know.
>
>
> Thank you very much,\
> Authors

---

> > ### Comment · Reviewer_jiih · 2025-11-27
> >
> > I appreciate the updated proof section in the paper. The newly added experiments address most of my concerns. Therefore, I am maintaining my score of 6 and increasing my confidence level to 4. The following are just a few minor questions I have about the updated paper; I would be grateful if the authors could answer them.
> >
> > 1. The paper repeatedly describes the reference distribution as
> > “**maximum uncertainty while remaining aligned with the target task distribution**.”
> > However, the only formal result provided is the monotonic relationship between softmax temperature and entropy (Appendix E). There is no formal definition or proof of “maximum uncertainty.” Could the authors **formally define** what “maximum uncertainty” means in this context (e.g., in terms of entropy over some constrained set of distributions)? Under which assumptions or constraints is the proposed reference distribution actually **maximizing** this uncertainty measure? As it stands, the theorem only shows that, for fixed logits, increasing temperature increases entropy. How do the authors justify the stronger statement that the proposed reference distribution represents “maximum uncertainty”? Should the wording be softened, or can a stronger formal argument be added?
> >
> > 2. The paper claims that the distribution obtained by masking text/state is still “aligned with the target task distribution” and is more reasonable than a uniform distribution. Can the authors provide any formal or empirical evidence (e.g., KL divergence, JS divergence, or other distance metrics) that the masked distribution is indeed closer to the full-condition distribution than, say, a uniform or arbitrary OOD distribution?

---

> ### Author Response · Authors · 2025-11-29
> **Response to Reviewer jiih**
>
> Dear reviewer jiih, \
> We sincerely appreciate your additional questions. We respond to each question in what follows.
>
> ---
> **[Q1] There is no formal definition or proof of “maximum uncertainty” for the reference distribution. Under which assumptions proposed reference distribution actually maximizes this uncertainty measure?**
>
> We clarify that “maximum uncertainty” refers to an action distribution (over action tokens $a$) that is maximally agnostic to the specific non-visual condition $M$, while still being constrained by the remaining condition $X$. We formalize this notion by defining, for fixed $x$, the optimal reference distribution $Q^{\*}(a)$ as the one that is unbiased with respect to any specific $m$ and is obtained by minimizing the expected KL divergence across all plausible conditioned distributions: $Q^{\*}(a)= \underset{Q}{\arg\min} \ \mathbb{E}_{m \sim p(m | x)} \ [\mathrm{KL}( p(a|x,m) || Q(a) )]$.
>
> We assume that the specific condition $m$ is sampled from the conditional distribution $p(m | x)$, for example, plausible task instructions are constrained by the observation in robotic manipulation. Expanding the expression gives $\mathbb{E}_{m} \ [ \mathrm{KL}( p(a | x,m) || Q(a) ) ] = -H(A| X,M) + H(p(a | x), Q(a))$, where the first term is constant with respect to $Q$. Thus, minimizing the objective reduces to minimizing the cross-entropy $H(p(a | x), Q(a))$. This cross-entropy term can be further decomposed as $H(p(a | x), Q(a)) = H(p(a | x)) + \mathrm{KL}( p(a | x) || Q(a) )$, which is minimized only when the KL term is zero (since KL is always non-negative). Therefore, we obtain $Q^{*}(a) = p(a | x)$. In conclusion, our condition-masking distribution serves as a reference distribution that is maximally agnostic to $M$ and therefore represents maximum uncertainty induced by the absence of $M$. Nevertheless, we acknowledge that the term “maximum uncertainty” may be misinterpreted as suggesting complete randomness (e.g., a uniform distribution), and we have softened the wording to “action uncertainty” in our revised manuscript, highlighted in “cyan”.
>
>
> ---
> **[Q2] Is there any formal or empirical evidence that the masked distribution is closer to the full-condition distribution than a uniform or other OOD distribution?**
>
> We clarify that our masking distribution remains close to the full-condition distribution because it is generated by the same VLA while preserving the visual context. To quantify alignment, we compute the average token-level KL divergence between the reference distribution $Q$ and the full-condition distribution $P$, $\mathrm{KL}(Q||P)$, under our test-time scaling framework. In the table below, first, the state-masking distribution yields substantially smaller KL divergence than the uniform distribution,  indicating that the masking distribution is far more aligned with the full-condition distribution. Second, state-masking with joint imitation learning achieves a lower KL than its counterpart without joint imitation learning, demonstrating that our joint training strategy improves the model's awareness of the masking distribution. Third, state-masking without joint imitation learning has significantly smaller KL than image-masking, showing that masking visual observations removes critical information required for task success. For your information, we report results from two models, one trained with imitation learning and the other trained with joint imitation learning using state-masking data, on SIMPLER “Carrot on Plate” task. Both models use the same configurations as in our main experiments (see Table 2), except that the regularization temperature of all masking distributions is fixed at 1 for clarity.
>
> \begin{array}{lc} \hline \text{Ref. Distribution $Q$}  & \text{Avg $D_{\mathrm{KL}}$ per token ($\downarrow$)} \newline \hline
> \text{State-masking (w/ Joint-IL)} & \textbf{30.17} \newline
> \text{State-masking (w/o Joint-IL)} & 33.02 \newline
> \text{Image-masking (w/o Joint-IL)} & 104.94 \newline
> \text{Uniform} & 148.78  \newline \hline
> \end{array}

---

### Author Response · Authors · 2025-11-22
**General Response**

Dear reviewers and AC,

We sincerely appreciate your valuable time and effort spent reviewing our manuscript.

As reviewers highlighted, our work introduces a verifier-free test-time scaling paradigm (All Reviewers), incorporates a theoretically motivated KL-based confidence measure (jiih), achieves gains in low-data and OOD settings (wMxK), provides comprehensive ablation studies (6QZo, CKBD), and is supported by well-structured and clear writing (CKBD).

We appreciate your constructive comments on our manuscript. In response to the comments, we have carefully revised and enhanced the manuscript with the following additional discussions and experiments:


- Editorial comments (Abstract, Introduction, Conclusion)
- Extended ablation study of number of candidates and aggregation strategies (Table 5b, Table 5f)
- Efficient deployment strategy (Figure 3, Appendix C)
- Ablation study of masking ratio on RoboCasa (Appendix D.1)
- Ablation study of aggregation strategies on SIMPLER-WidowX (Appendix D.2)
- Effect of model scale on LIBERO (Appendix D.3)
- Comparison with external-verifier method on SIMPLER-WidowX (Appendix D.4)
- Theoretical justification of regularization temperature in reference distribution (Appendix E)
- CALVIN benchmark results (Appendix F)
- Visualizations for failure cases of MG-Select (Appendix G)
- Visualizations for token-level KL divergence of MG-Select (Appendix G)
- Analysis of truncated FAST tokens (Appendix H)

In the revised manuscript, these updates are temporarily highlighted in “blue” for your convenience to check.

We hope our response and revision sincerely address all the reviewers’ concerns.

Thank you very much,\
Authors.

---

### Public Comment · ~Suhyeok_Jang1 · 2026-03-02
**Correction to Table 6**

We identified that the reported numbers of OpenVLA in Table 6 were slightly incorrect due to an aggregation mistake during task-level result summarization. In the camera-ready revision, we have corrected these values without any re-training or re-evaluation. We note that the overall trend remains exactly the same, and this correction does not affect the conclusions of the paper.

---

### Meta-Review · Area_Chair_zqZu · 2026-01-07

**Summary:**

The paper proposes MG-Select, a method for test-time scaling of VLA models without requiring external verifiers or additional training. The core idea is to sample multiple action candidates and select the best one based on a confidence metric.  The authors also introduce a joint training strategy with condition dropout to improve the quality of this reference distribution.

The review process was positive, with a unanimous consensus for acceptance. Reviewers appreciated the verifier-free paradigm, which addresses an important bottleneck in deploying large VLA models by removing the need for costly external reward models. The method's simplicity and its consistent empirical gains across diverse benchmarks (RoboCasa, LIBERO, SIMPLER) were highlighted as key strengths. Initial concerns regarding the theoretical justification of the metric and the method's applicability to long-horizon tasks were mostly addressed during the rebuttal.

**Reviewer Concerns:**

- Reviewers CKBD and 6QZo initially questioned the formal justification of the KL divergence confidence metric, concerned it might be merely an empirical heuristic without theoretical backing.

- A major limitation noted was the exclusive focus on short-horizon pick-and-place tasks, raising doubts about the method's applicability to long-horizon, multi-step planning scenarios.

- Reviewers queried the method's robustness across different model scales and different tokenization architectures.

- There were requests for direct comparisons with external verifier methods to validate claims regarding efficiency and performance trade-offs.

The authors comprehensively addressed all the major concerns raised during the rebuttal. No major concerns remained outstanding.

**Reviewer Scores:**

The scores converged to a unanimous 6. Reviewer CKBD explicitly raised their score from 4 to 6 after the rebuttal. Reviewer jiih also increased their confidence.

---

### Decision · Program_Chairs · 2026-01-26

Accept (Poster)